



# Mantle flow under the Central Alps: Constraints from non-vertical SKS shear-wave splitting

Eric Löberich[1] and Götz Bokelmann[1]

[1]Department of Meteorology and Geophysics, University of Vienna, Vienna, 1090, Austria

**Correspondence:** Eric Löberich (eric.loeberich@univie.ac.at)

**Abstract.** The association of seismic anisotropy and deformation, as e.g. exploited by shear-wave splitting measurements, provides a unique opportunity to map the orientation of geodynamic processes in the upper mantle and to constraint their nature. However, due to the limited depth-resolution of steeply arriving core-phases, used for shear-wave splitting investigations, it appears difficult to differentiate between asthenospheric and lithospheric origins of observed seismic anisotropy. To change
that, we take advantage of the different backazimuthal variations of fast orientation $\phi$ and delay time $\Delta t$, when considering the non-vertical incidence of phases passing through an olivine block with vertical b-axis as opposed to one with vertical c-axis. Both these alignments can occur depending on the type of deformation, e.g. a sub-horizontal foliation orientation in the case of a simple asthenospheric flow and a sub-vertical foliation when considering vertically-coherent deformation in the lithosphere. In this study we investigate the cause of seismic anisotropy in the Central Alps. Combining high-quality shear-wave
splitting measurements of three datasets leads to a dense station coverage. Fast orientations $\phi$ show a spatially coherent and relatively simple mountain-chain-parallel pattern, likely related to a single-layer case of upper mantle anisotropy. Considering the measurements of the whole study area together, our non-vertical-ray shear-wave splitting procedure points towards a b-up olivine situation and thus favors an asthenospheric anisotropy source, with a horizontal flow plane of deformation. We also test the influence of position relative to the European slab, distinguishing a northern and southern subarea based on vertically-
integrated travel times through a tomographic model. Differences in the statistical distribution of splitting parameters $\phi$ and $\Delta t$, and in the backazimuthal variation of $\delta\phi$ and $\delta\Delta t$, become apparent. While the observed seismic anisotropy in the northern subarea shows indications of asthenospheric flow, likely a depth-dependent plane Couette-Poiseuille flow around the Alps, the origin in the southern subarea remains more difficult to determine and may also contain effects from the slab itself.





## 1 Introduction

The propagation of seismic waves is affected by the properties of each layer they pass, providing the opportunity to deduce information about interior structures and dynamics of Earth. The observation of directionally-dependent phase velocities, called seismic anisotropy, is such a property, which can be related to e.g. crystal alignment in upper mantle minerals, referred to as lattice-preferred orientation (LPO) (see e.g. Mainprice, 2015). Olivine is known to be the most common anisotropic mineral in this depth domain (see e.g. McDonough and Rudnick, 1998) and the fabric that it controls can have several types, depending on water content, temperature, and pressure conditions (Karato, 2008). Additionally, other minerals like orthopyroxene, clinopyroxene, and garnet also show anisotropy, and they effect the observations (Nicolas and Christensen, 1987; Babuska and Cara, 1991; Karato, 2008; Mainprice, 2015, and references therein).

Following Vinnik et al. (1984) and Silver and Chan (1988) the occurrence of shear-wave splitting (SWS) is one of the observations that reveal seismic anisotropy. Shear-wave splitting is comparable to the phenomenon of optical birefringence and it describes that an S-wave splits into two waves, when it passes through an anisotropic layer – forming waves with a slow and a fast polarization orientation $\phi$. The polarization of both quasi-S-waves (qS) is perpendicular to each other. The delay $\delta t$ between the qS-waves accumulates depending on the strength of anisotropy (based on the variation of phase velocities) and the length of the path through the medium.

The presence of receiver-side anisotropy shows up as elliptical particle motion when observing core phases, like SKS and SKKS, of stronger earthquakes ($M > 5.5$) in a certain distance ($\sim 80°$ - $120°$). Other than in the case of an isotropic Earth, not only the radial-, but also transverse component shows a signal. Using e.g. the minimum-energy technique of Silver and Chan (1991), the SWS parameters $\phi$ and $\Delta t$ can be determined (see also Vinnik et al., 1989) applying a grid search to linearize the observed particle motion and thus removing the effect of anisotropy (see e.g. Wüstefeld et al., 2008).

Since the observed $\phi$-$\Delta t$ pattern reflects LPO anisotropy, SWS measurements hold the potential to relate mineral alignment, subsurface deformation, and geodynamic settings across scales, but due to the near-vertical arrival of core phases, the method has only weak depth resolution. So the orientation of the rock foliation could not been derived for a long time, what was particularly unfortunate as it would suggest a flow-plane orientation (Bokelmann, 2002a, 2002b and references therein) and thus further restrict the cause of anisotropy. Whether the observed SWS pattern represents more recent flow activity in the asthenosphere (Vinnik et al., 1989; Savage and Silver, 1993) or is likely related to ancient deformation frozen-in lithosphere (Silver and Chan, 1988, 1991), has thus been under debate. In Löberich and Bokelmann (2020) we have developed a procedure to relate differences in the small-scale azimuthal variations in $\phi$ and $\Delta t$ to foliation orientation and subsurface deformation, taking advantage of the non-vertical arrival of SKS phases, assuming a dominant single-layer case of seismic anisotropy. Investigating a big dataset of automatically determined SWS measurements (Liu et al., 2014), we were able to determine the source of seismic anisotropy beneath Western/Central US as a mainly subhorizontally-oriented flow plane of deformation.





Such a density and amount of stations and measurements are indeed unique for SWS studies, but using a reasonable number of SWS measurements of higher precision, as available e.g. in the Central Alps, yields another opportunity to test our approach.

In this study we demonstrate how the analysis of non-vertical SWS can help to constrain the cause of seismic anisotropy

below the Central Alps, improving our knowledge on deformation mainly in the European upper mantle. Combining previous SWS measurements in the Alps (Barruol et al., 2011; Qorbani et al., 2015; Salimbeni et al., 2018), the fast orientations $\phi$ reveal a spatially consistent pattern, usually described as being parallel to the mountain-chain (see e.g. Bokelmann et al., 2013), and related to a single-layer case of seismic anisotropy in the Western and Central Alps. However, the Eastern Alps show evidence for a double-layer case possibly related to the presence of a slab detachment (Qorbani et al., 2015). Even if the Western and

Central Alpine region seems to be less complex in terms of SWS, our understanding of tectonic settings and related geodynamic processes is still incomplete or even contradicting (see e.g. Kästle et al., 2019).

Subsequently we investigate the backazimuthal variations of SWS parameters in the whole Central Alps study area simultaneously and in a northern and southern subarea separately, to examine whether their distributions can be related to a Simple

Asthenospheric Flow (SAF) or Vertically-Coherent Deformation (VCD). In this context we also compare the SWS pattern with the $dVp$ model of Koulakov et al. (2009) to determine whether the behavior of SWS measurements, in relation with the different $dVp$ polarities in upper mantle depths, favor different causes of seismic anisotropy in the two subareas.





## 2 Tectonic Setting

The history of Alpine orogeny, leading to the arc-like mountain-chain (Fig. 1) known today, was controlled by the convergence of Eurasia and Africa, two major continental plates; it was further defined by related opening and closing processes of oceanic domains (see e.g. Coward and Dietrich, 1989). Following Handy et al. (2010), Neotethys with its subbranches (Io-

nian Sea and Meliata-Maliac-Varder) opened during the late Paleozoic-Mesozoic, while Pangea broke apart. However, in the Jurassic-Cretaceous both branches have been reworked during subduction and obduction processes (Handy et al., 2010, based on Stampfli et al., 1998; Schmid et al., 2008; Faupl and Wagreich, 2000), while the Alpine Tethys (Handy et al., 2010, based on the naming in Stampfli and Borel, 2002) and Atlantic Ocean opened jointly. Eventually, the Alpine Tethys formed upon two ocean basins, namely Piemont-Ligurian and Valais, separated by the Briançonnais terrain (for further details see e.g. Stampfli,

1994, based on Stampfli, 1993). When the European Plate underwent subduction below Adria, a microplate assumed to be an African promontory (Handy et al., 2010, based on Argand, 1924; Channell and Horváth, 1976; Channell et al., 1979), these entrapped basins closed successively, starting with Ligurian segments (131-118 Ma), followed by Piemont, and further Western Liguria and Valais (84-35 Ma). This process was completed by the incipient Alpine collision 35 Ma ago (see e.g. Handy et al., 2010). Following e.g. Schlunegger and Kissling (2015) (based on Schmid et al., 1996) the change from the initially

subducted southern oceanic rim to continental lithosphere of the European plate, caused differential forces in the subduction system (Davies and Von Blanckenburg, 1995) what finally led to a slab break-off of the oceanic section (Dal Piaz and Gosso, 1994; Von Blanckenburg and Davies, 1995) and further an asthenospheric upwelling. As consequence, exhumation took place till 20 Ma (Schlunegger and Kissling, 2015, based on Pfiffner, 1986; Schlunegger et al., 1997; Kempf et al., 1999), followed by a northward movement of the Apenninic slab (Salimbeni et al., 2018, based on Malusà et al., 2015; Malusà et al., 2016).

However, the slab break-off occurrence time and subsequent movement in each Alpine section are still under debate (Handy et al., 2010, and references therein).

Nowadays we assume the Alpine mountain range to be built upon a two-lithosphere root system, namely the slabs situated under the Western/Central Alps and Eastern Alps (see e.g. Qorbani et al., 2015, based on tomographic models of Lippitsch

et al.,2003; Koulakov et al., 2009; Mitterbauer et al., 2011). This geodynamic situation is already complex, but the appearance of the surrounding Appeninnic, Carpathian, and Dinaridic slabs, dipping southwest-, east-, and northeastwards (see e.g. Handy et al., 2010 or Salimbeni et al., 2018, based on Lucente et al., 1999; Piromallo and Morelli, 2003; Giacomuzzi et al., 2011; Zhao et al., 2016; Hua et al., 2017), shows the overall tectonic diversity and related activity, e.g. the ongoing consumption of Adria, in the Mediterranean region (Handy et al., 2010). The variety of subsystems of the Central Mediterranean tectonics

makes it difficult to derive a unique model from geophysical measurements capable to explain and relate all geodynamic implications, as e.g. shown by Kästle et al. (2019) comparing findings of different tomographic models of Lippitsch et al. (2003); Koulakov et al. (2009); Dando et al. (2011); Mitterbauer et al. (2011); Zhao et al. (2016); Hua et al. (2017); Kästle et al. (2018).

Beside the velocity structures obtained from different inversion methods of various wave types and phases investigated at





a certain time, seismic anisotropy and particularly SWS measurements hold the potential to complete our basic (model-like) understanding of upper mantle processes, e.g. by taking advantage of the relation between the direction of accumulated deformation and observed fast orientation $\phi$. Barruol et al. (2011) stated that the gradual clockwise rotation of $\phi$ along the Alpine belt indicates a curvature of the geometry of internal deformation. Further, the delay time $\Delta t$ provides quantitative information

about the thickness and strength of an anisotropic layer. However, internal Alpine regions surprisingly unveil relatively weak anisotropy, although this area probably underwent higher rates of deformation than external regions, which show stronger anisotropy. This previous work already indicated that the Western Alps SWS pattern could be caused by an asthenospheric source, discussing a passive flow, shaped by a European slab keel, following the absolute plate motion of Eurasia. Other possibilities are described by an active flow, e.g. generated by a European slab detachment (see e.g. Lippitsch et al., 2003) or due

to the influence of the European and Appeninic slab rollbacks upon the flow system (Barruol et al., 2011, based on Funiciello et al., 2006; Piromallo et al., 2006; Vignaroli et al., 2008).

Recently Salimbeni et al. (2018) provided additional SWS measurements in Western Alps and Apennines, spatially densifying the general mountain-chain-parallel pattern of $\phi$. Contrary to Barruol et al. (2011), this study excluded the approach of

an active flow, driven by a discontinuous European slab, based on the recent tomography findings of Zhao et al. (2016). It disagreed with the possibility of a European slab rollback (Salimbeni et al., 2018, based on Handy et al., 2010; Malusà et al., 2015) and further reinterpreted the influence of the Apenninic slab rollback as follows. Based on the model of Zhao et al. (2016), the European and Apenninic slabs seem to be connected. Hence a toroidal flow, as would have been generated by the Apenninic slab rollback, could not form. This led to a mass deficit west of the Apenninic slab. To overcome this imbalance, it

is assumed that asthenospheric material was pulled behind from western and northern areas, generating a counterflow, shaped according to the European slab appearance. This may include small-scale upwelling (Salimbeni et al., 2018, based on Long and Becker, 2010; Díaz et al., 2013), which would evoke an upper-mantle temperature anomaly and might facilitate topographic uplift (see e.g. Salimbeni et al., 2018, based on Chéry et al., 2016; Nocquet et al., 2016 or Sternai et al., 2019, and references therein) in agreement with the highest mountains being generally situated in the Western Alps.

Beside the asthenospheric cause of SWS, Salimbeni et al. (2018) (and references therein) also pointed out the potential of additional lithospheric (slab-related) anisotropy, so called fossil fabric, based on frozen-in deformation from Tethyan rifting. Such an additional lithospheric source of anisotropy has been found in the Eastern Alps. While the gradual clockwise rotation characteristic of $\phi$ remains preserved (see e.g. Bokelmann et al., 2013), Qorbani et al. (2015) observed a $\pi/2$-periodicity in the

Eastern Alps, consistent with a double-layer anisotropy case. Here the upper layer is characterized by a NW-SE fast orientation, possibly linked to asthenospheric flow activity, contrary to the deeper layer, which unveils a generally NE-SW pattern associable with a European slab detachment. Whether the observed pattern of SWS shows a dominant asthenospheric or lithospheric origin is often difficult to distinguish, but following the non-vertical SWS procedure we are now able to further constrain the cause of seismic anisotropy.





## 3 Method and Data

As introduced in detail in Löberich and Bokelmann (2020) our approach is based on a Taylor-series expansion, derived by Davis (2003) for a horizontally-oriented single-layer case of (orthorhombic) seismic anisotropy, which describes the azimuthal behavior of both SWS parameters in the case of non-vertical SKS phase arrival (incidence: $\theta \leq 30°$). Eq. 1-2 show the related

solution of the fast orientation $\phi$ as sum of the vertical incidence case $\phi_0$ and an additional term of small-amplitude oscillation $\delta\phi$:

$$\phi = \phi_0 + \delta\phi, \tag{1}$$

with

$$\delta\phi = d_1 \sin(2z)\theta^2. \tag{2}$$

This variation introduces angular dependencies, namely a $180°$-periodicity on azimuth $z$ with amplitudes being controlled by the incidence $\theta$. The effect of the medium and its orientation, as represented by the stiffness tensor $C$, is expressed by $d_1$ as:

$$d_1 = -f1/f4, \tag{3}$$

with

$$f_1 = C_{1212} - C_{2233} - C_{1133} - 2C_{1313} + C_{1122} - C_{2323} + C_{3333}, \text{ and } f_4 = -2C_{1313} + 2C_{2323}. \tag{4}$$

Since $d_1$ specifies the phase polarity we subsequently refer it as "oscillation parameter". A similar expression can be derived for the delay time $\Delta t$ in the case of non-vertical incidence, where:

$$\Delta t = \Delta t_0 + e_1\theta^2 + \delta\Delta t, \tag{5}$$

with

$$\delta\Delta t = e_2 \cos(2z)\theta^2. \tag{6}$$

Then

$$e_1 = \frac{D}{2}\sqrt{\frac{\rho}{\bar{c}^3}}\,(F_1 - S_1)\,, \text{ and } e_2 = \frac{D}{2}\sqrt{\frac{\rho}{\bar{c}^3}}\,(F_2 - S_2) \tag{7}$$

with $\rho$ being the density, $D$ specifying the path length,

$$F_1 = -\frac{5}{2}C_{1313} - C_{1133} + \frac{1}{2}C_{1111} + \frac{1}{2}C_{3333} + \frac{1}{2}C_{1212}\,,$$
$$F_2 = -\frac{3}{2}C_{1313} - C_{1133} + \frac{1}{2}C_{1111} + \frac{1}{2}C_{3333} - \frac{1}{2}C_{1212}\,,$$
$$S_1 = -\frac{5}{2}C_{2323} - C_{2233} + \frac{1}{2}C_{2222} + \frac{1}{2}C_{3333} + \frac{1}{2}C_{1212}\,,$$
$$S_2 = \frac{3}{2}C_{2323} + C_{2233} - \frac{1}{2}C_{2222} - \frac{1}{2}C_{3333} + \frac{1}{2}C_{1212}\,,$$
$$\text{and } \bar{c} = \sqrt{C_{1313}C_{2323}}\,. \tag{8}$$



In Löberich and Bokelmann (2020) we have shown that $d_1$ and $e_2$ vary for different orientations of olivine. Assuming a horizontal a-axis (Nicolas and Christensen, 1987) together with a vertically-oriented b-axis (b-up) or c-axis (c-up) changes the amplitude of $\delta\Delta t$ due to $e_2$ alteration and shifts the phase in $\delta\phi$, meaning opposite polarities of $d_1$ (Fig. 2). While a b-up orientation leads to a negative $d_1$ value, a c-up alignment causes a positive $d_1$. Hence, assuming a broad backazimuthal distribution

5    of high-quality individuals SWS measurements, both cases are indeed distinguishable especially due to the difference in the oscillation parameter. This is particularly interesting as these orientations can be related to a SAF and VCD scenario, mentioned before. Silver (1996), and references therein, summarized that anisotropy, caused by the asthenosphere, is associated with a b-up olivine orientation, assuming a horizontal flow plane of deformation and foliation orientation, and lithospheric anisotropy, related to frozen-in deformation and vertical foliation, might lead to a c-up olivine alignment.

To distinguish both cases, we investigate only good quality individual SWS measurements of SK(K)S phases determined using the minimum energy technique (Silver and Chan, 1991) by Barruol et al. (2011) (SDSN/CH, 2006 - 2008, $\Delta$: $85°$ - $120°$, $M > 6$), Qorbani et al. (2015) (OE, 2002 - 2013 $\Delta$: $90°$ - $130°$, $M > 6$) and Salimbeni et al. (2018) (IV, 2012 - 2013, $\Delta$: $80°$ - $120°$, $M > 5.8$). We further restrict our selection introducing a threshold on their standard deviation $\sigma_\phi$ of $20°$, to

15    ensure the fast orientation measurements in our dataset have a similar precision. The SWS pattern related to those high-quality measurements in the study area can be broadly described as mountain-chain-parallel (Fig. 3 top). Later we will show that the observed pattern is actually more complex. The fast orientations (bottom, left) in general rotate clockwise from NNE-SSW in the west to NE-SW in the east, mainly pointing towards the NE-SW orientation. Delay times (bottom, right) range between $\sim 0.48$ - $2.88\,\text{s}$, with an accumulation between $\sim 0.96$ - $1.92\,\text{s}$.





## 4 Results

The variations $\delta\phi$ and $\delta\Delta t$ of the observed SWS parameters provide a possibility to constrain $d_1$ and $e_2$, as shown in Eqs. 2 and 6. To derive $\delta\phi$ and $\delta\Delta t$, we follow the workflow described in Löberich and Bokelmann (2020). Fig. 4 (top row) shows the backazimuthal distribution of the high-quality SWS data with $\phi$ in the uniform interval $[0°:180°]$. As also recognizable

from the map in Fig. 3 the measurements accumulate for backazimuths of $45°$ - $90°$ and $180°$ - $315°$. Since earthquakes do not occur at every azimuth the distribution itself is biased, missing information e.g. from north and southeast direction. Furthermore, SWS parameters cannot be obtained for backazimuths parallel or perpendicular to $\phi$ and lead to so called Null measurements. For the subsequent investigation we only consider stations with more than five measurements to ensure that the means per station, $\overline{\phi}$ and $\overline{\Delta t}$, are determined stably. The backazimuthal distribution of the SWS parameters is then cor-

rected for the corresponding $\overline{\phi}$ and $\overline{\Delta t}$, which should align the measurements as if they would have been recorded at a single station (second row). This step renders the quantities $\delta\phi$ and $\delta\Delta t$ as in Eqs. 2 and 6. Higher complexities of anisotropy could in principle bias the calculation of the mean, but the consistent simplicity of the SWS pattern in the Central Alps let them be unlikely and instead favor a single-layer case of anisotropy. To avoid effects of possible outliers, we introduce thresholds based on both data distributions ($5\%$ and $90\%$ quantile). We further select only measurements in intervals ($\pm33.25°$) around

backazimuths of $45°$, $135°$, etc., where the expected difference between b-up and c-up olivine variation in $\delta\phi$ is the largest (adapted from gray areas in Fig. 2); from these we determine mean and median of $\delta\phi$ and $\delta\Delta t$ (third row). As the amount of data is much less than in our previous study on the Western and Central US, the thresholds and intervals are adapted and chosen less restrictive. Even though the meaningfulness of the $135°$ and $315°$ interval is reduced due to lower data density, a backazimuthal variation of the means and medians of $\delta\phi$ is still recognizable, while the equivalent $\delta\Delta t$ values are, as expected,

nearly 0 in the investigated intervals. The comparison of means and medians of $\delta\phi$ (and $\delta\Delta t$) with the derived variations from the forward modelling (bottom row), in which the related error of the mean is based on the double standard deviation $\sigma$ and weighted by the square root of the number of datapoints per interval, suggest a b-up olivine situation in the study area and thus a horizontal flow plane of deformation as described by the SAF model. Assuming a dominant source of flow-related anisotropy, is in agreement with the model expectations of Barruol et al. (2011) and Salimbeni et al. (2018). However, investigating the

whole area at once might be too simplistic as the tectonic settings across the subduction zone are complex. The verification of our outcome can be achieved by a comparison with tomographic results of the region.

Fig. 5 shows a comparison of the high-quality individual SWS measurements and the related portion of the $dVp$ model of Koulakov et al. (2009). Slicing the model at $150\,\mathrm{km}$ depth (top) reveals a lateral transition from a low- to high-velocity zone,

likely related to the continuation of the counterflow (see e.g. Salimbeni et al., 2018) around the Western Alps and the occurrence of the European slab, respectively. The SWS pattern, usually described as mountain-chain-parallel, can apparently be separated, into the low- and the high-velocity anomaly. For each of the two anomalies the fast orientation seems to follow the shape of the anomaly. To understand this behavior we investigate three $dVp$ depth profiles through the wider area.





Profile A (second row, left) slices the region diagonally from northwest to southeast, crossing the low- and high-velocity anomalies mentioned before. Considering the main part of both regions, they show a comparable depth extent. While the low-velocity anomaly is situated at depths of $\sim 80$ - $260\,\mathrm{km}$ and thus extends $180\,\mathrm{km}$ vertically, the high-velocity anomaly has a slightly smaller vertical extent ($150\,\mathrm{km}$); it is between $\sim 90$ - $240\,\mathrm{km}$ depth. Both are separated by a sharp nearly-vertical

transition (see e.g. Koulakov et al., 2009) at a profile distance of $\sim 290\,\mathrm{km}$. However, between $\sim 240$ - $290\,\mathrm{km}$ the slab thins massively and starting at $\sim 250\,\mathrm{km}$ gets overlain by a strong low-$dVp$ Po plain anomaly (see e.g. Lippitsch et al., 2003, based on Spakman et al., 1993; Solarino et al., 1996; Bijwaard and Spakman, 2000 or Koulakov et al., 2009), which has been argued to reveal a hydrated Adriatic mantle wedge (see e.g. Giacomuzzi et al., 2011, and for further reading Hearn, 1999). Similarly as in the tomographic image also $\phi$ (third row, left) shows a transition from NE to ENE along profile A between $260$ - $310\,\mathrm{km}$,

while $\Delta t$ values (bottom, left) are typically between $\sim 1$ - $2\,\mathrm{s}$ without showing a clear change when crossing both structures. If we assume that the strength of LPO anisotropy is comparable throughout the study region, the path length D through the structures causing SWS should be also similar. This points at a different origin of the anisotropy in the north and the south. If the low-velocity block in the north is giving rise to the (flow-related) anisotropy there, as seen in the non-vertical SWS analysis, then it should be the fast-velocity anomaly in the south, giving rise to lithospheric anisotropy (see below).

While profile A crosses both structures, B and C are aligned with both anomalies, respectively. Profile B (second row, second column) follows the low-velocity zone from southwest to northeast, revealing a slightly dipping structure of $180$ - $200\,\mathrm{km}$ depth extension, situated between depths of $\sim 160$ - $340\,\mathrm{km}$ in the southwest, and $\sim 80$ - $280\mathrm{km}$ in the northeast. The fast orientation (third row, second column) continuously follows the low-velocity zone, rotating from N (outside the study area) towards ENE, with delay times (bottom, second column) mainly varying between $\sim 0.5$ - $2\,\mathrm{s}$. Profile C is oriented south to

east, parallel to the high-velocity zone, which shows a depth extent of $230\,\mathrm{km}$, between depths of $\sim 80$ - $310\,\mathrm{km}$ in the south and $\sim 90$ - $270\,\mathrm{km}$ in the east. However, the tomography shows strong variations along the profile, particularly at a distance of $\sim 270\,\mathrm{km}$. The weakening to $40\,\mathrm{km}$ vertical extent (depths of $\sim 130$ - $170\,\mathrm{km}$) suggests a shortening of the slab in the contact zone between the Western/Central and Eastern European slab. The fast orientations (third row, right) constantly point towards ENE following the striking of the high-velocity zone. Further the delay times (bottom, right) at stations of Barruol et al. (2011)

and Qorbani et al. (2015) also show indications of the shortening of the slab. While $\Delta t$ mainly varies between $\sim 0.5$ - $2\,\mathrm{s}$ in the south and $\sim 1$ - $2\,\mathrm{s}$ in the east, it decreases down to $\sim 1.1\,\mathrm{s}$ and $\sim 0.7\,\mathrm{s}$ at $180\,\mathrm{km}$ and $210\,\mathrm{km}$ distance, giving further evidence for a lithospheric-related cause of anisotropy in this region. So far we have just compared the spatial pattern of high-quality SWS measurements with the $dVp$ distribution of Koulakov et al. (2009) at $150\,\mathrm{km}$ depth. Knowing the significant changes

in the velocity field of the upper mantle in the vicinity of the subduction zone, as seen e.g. in Profile A, encourages further analysis of the lateral variation.

Fig. 6 shows the depth dependence of the lateral variation of $dVp$ at $90\,\mathrm{km}$, $140\,\mathrm{km}$ and $190\,\mathrm{km}$ indicating that both areas, low- and high-velocity, get more stable with increasing upper mantle depth. The variation of the high-dVp strength along

strike for example is particularly obvious at $90\,\mathrm{km}$ depth, and it spatially correlates with the general $\Delta t$ changes, supporting





the approach of a lithospheric, slab-related anisotropy as stated before. Further, the geometry of the low-dVp anomaly in the north reveals a higher complexity at shallower upper mantle depth (e.g. $90\,\mathrm{km}$) than at larger depth. If real, then this could explain the backazimuthal dependence of $\phi$ at a station around $46°45'\mathrm{N}\,8°\mathrm{E}$, where SWS measurements from earthquakes in the northeast to east direction in general lead to a NE-SW fast orientation and measurements related to earthquakes from the
southwest to west point more towards NNE-SSW. Since the slab below this station might be weakened, according to profile A, the dominant source of seismic anisotropy should be related to the flow orientation in the asthenosphere. Following the model of Salimbeni et al. (2018), this counterflow compensates the mass deficit, generated by the slab rollback of the Apennines, and thus attracts material to move around the Western/Central European slab. As this station is located at the edge of the slab in $90\,\mathrm{km}$ depth, the flow itself can be expected to become more complex. Hence, the backazimuthal difference in $\phi$ might indicate
two branches of a depth-dependent flow system. While ray paths of both backazimuthal directions get influenced by the mainly NE-SW oriented flow at greater depths, only earthquakes from the southwest to west might image the shallower flow, turning around the slab edge in a NNE-SSW direction, as ray paths of earthquakes from northeast to east are likely shielded by the slab.

Fig. 5 and 6 indicated that the SWS pattern rather correlates with the extent of the low- and high-$dVp$ zone than to be com-
pletely mountain-chain-parallel, which suggests a separation into subareas to test the possibility of different causes of seismic anisotropy. To take the $dVp$ depth variation into account, we calculate vertically-integrated travel times through the model of Koulakov et al. (2009) at upper mantle depths (77.5 - $410\,\mathrm{km}$) and use the lateral variation as criterion to define subareas in Fig. 7 (top). To ensure our distinction is not biased by the specific model, we also determine travel times for the model of Hua et al. (2017) in Appendix A. In general, the spatial distribution of travel times in Fig. 7 (top) looks similar like the behavior of
$dVp$ seen before at e.g. $140\,\mathrm{km}$, $150\,\mathrm{km}$ and $190\,\mathrm{km}$, confirming that these distributions of velocity perturbations are relatively stable and representative across the upper mantle. Overall the northern subarea is characterized by longer travel times, due to the lower velocities of the asthenospheric flow below, and a rotation of $\phi$ along the Alps, while in the southern subarea of shorter travel times, related to the high velocities of the lithospheric slab, $\phi$ seem to be parallel to the $dVp$ anomaly, showing only little rotation. Considering the statistical distribution of the SWS parameters, the fast orientations of the northern subarea
(second row, left) follow a similar distribution as seen before for the whole study area. Further, the delay times (second row, right) accumulate between $\sim 1.28$ - $1.92\,\mathrm{s}$ with a sharp transition to lower and higher $\Delta t$ values. In comparison to the northern subarea, $\phi$ is mainly oriented towards NE-SW in the southern subarea (bottom, left), and $\Delta t$ (bottom, right) does not tend to higher values (rather between $\sim 0.96$ - $1.76\,\mathrm{s}$). This changing behavior further supports the idea of two different causes of the observed SWS pattern.

Taking advantage of the spatial subdivision of high-quality SWS measurements, we follow the procedure as explained for the whole study area to investigate the backazimuthal variation of $\delta\phi$ and $\delta\Delta t$ for the northern and southern subarea separately (Fig. 8 and 9). Both subsets show comparable parameter distributions (top row), however the northern subarea is better constrained due to the higher number of measurements. Applying the same criteria for thresholds and intervals (second row)
as before, reveals a similar behavior for $\delta\phi$ and $\delta\Delta t$ (third row) in the northern subarea as seen for the whole study region.





This suggests an olivine b-up orientation related to an asthenospheric cause of anisotropy (bottom row). However, due to the reduced amount of datapoints, e.g. at $135\,^\circ$, the uniqueness of the distinction gets lost ($2\sigma$ error) and the significance is reduced. In the southern subarea the significance of most intervals is lower due to the reduced coverage in the selected, $\overline{\phi}$-compensated backazimuthal distribution (third row). A stable differentiation between different olivine orientation (bottom row), based on

5   the non-vertical SWS approach, is thus difficult with the amount of currently available high-quality SWS measurements in the subarea.





## 5 Discussion

The analysis of high-quality SKS splitting measurements in the Central Alps suggests that the SKS splitting indeed shows small but systematic variations with backazimuth. The angular dependence revealed is consistent with that expected from the dominant foliation orientation of anisotropic minerals in the upper mantle; it thus holds direct constraints on subsurface defor-

mation and the nature of the seismic anisotropy, which was ambiguous previously due to the weak depth resolution of SWS. Considering measurements in the entire study area at once, the variations agree with a simple model of olivine with a horizontal foliation and vertical b-axis (b-up olivine), just as the one indicated in Fig. 2. This kind of anisotropy is expected, if the LPO is generated by an asthenospheric flow around the Alps.

As the observed backazimuthal variation of splitting parameters is not large and requires a broader azimuthal distribution to become more apparent, it has not been studied before. In any case, the importance of an observation is not necessarily related to its size; if one had not considered small effects there would be no quantum mechanics, no gravitational waves etc. All that matters is whether the effect is significant, and this is the case, judging from the error bars in Fig. 4. Even a moderate number of "good" quality splitting measurements is apparently enough to constrain the effect, which suggests that such studies

can also be performed at regional scale.

Our analysis has assumed that the anisotropy is well-characterized by a single horizontal layer of anisotropy. Indeed, the Central Alps have been well-characterized by single-layer anisotropy before, and there are no signs of two-layer cases ($90°$ periodicity) or a dipping layer ($360°$ periodicity) in the area. In the latter case of dipping layer anisotropy the equations used

here could be further adjusted following Davis (2003). If there were more anisotropic layers, but only one would be dominant, this would not necessarily rule out our approach, but it would require more data and a wider range of backazimuths to stabilize the procedure (see Löberich and Bokelmann, 2020). Since the area of study is already quite well-covered with permanent broadband instruments, the presence of AlpArray does not provide many additional stations, but as the dataset so far only covers a relatively limited time duration it can in principle be extended for a longer time period.

As much as one might wish to be able to use a larger range of backazimuths, we must admit that near the Null directions, parallel or perpendicular to the fast orientation, the observations become less stable. Hence, we need to average over a backazimuthal window, which is set to $±33.25°$ width around maxima and minima of $δ\phi$ to distinguish SAF and VCD. This stabilizes the effect in $δ\phi$, but it has the adverse effect of rendering the constraint in $δ\Delta t$ less useful. However, the two endmember mod-

els can hardly be distinguished by the splitting delay variation anyway.

Considering the models themselves, it is remarkable that the SWS observations were fit by our simplistic starting configuration without adapting it. Since the actual amount of aligned olivine is not known, we assumed the approximately $70\%$ olivine of the upper mantle (see e.g. McDonough and Rudnick, 1998 or Faccenda and Capitanio, 2013) to be perfectly-aligned,





while $30\%$ remain randomly-oriented. However, we have shown in Löberich and Bokelmann (2020) that reasonable changes of this ratio do not significantly affect the $d_1$ parameter. To approximate the observations, it was also not necessary to consider other minerals like orthopyroxene (Nicolas and Christensen, 1987; Babuska and Cara, 1991; Karato, 2008; Mainprice, 2015, and references therein) or to involve the effect of e.g. pressure (depth) to make the model more realistic. So we used the Voigt-

Reuss-Hill average, as in our previous study in the Western and Central US, to model the behavior of seismic anisotropy, and it is striking, but probably not fortuitous, that the same b-up configuration can also explain the Central Alps data.

An intriguing consequence of the observations is that a girdle configuration of olivine grains, as would be implied by several deformation mechanisms (see e.g. Nicolas and Christensen, 1987), is clearly ruled out by the data, which rather favors

a "high-temperature mechanism". Otherwise we would obtain $d_1$ values in the vicinity of zero (Löberich and Bokelmann, 2020). We regard this as an important conclusion, since petrologists often assume such girdle configurations, based on findings of individual xenoliths (see e.g. Soustelle et al., 2010). However, the latter give rather localized information and may not be representative of larger regions of the upper mantle.

Considering smaller regions, we found indications that the backazimuthal variations in the northern subarea are also consistent with the predictions for a horizontal alignment of foliation planes, as expected from the SAF model. However, the reduced amount of measurements makes it more complicated to constrain the b-up situation here.

Assuming the source of anisotropy to be related to the presence of the horizontal upper mantle flow, seen along profile B

around the Alps (Fig. 10 left), we subsequently consider further inferences about the flow type. The SAF model used here is a first approximation to understand the flow field to some extent, but as seen in Fig. 6, shallower depth already indicated the possibility of a more complex, depth-dependent system. In general, there are two simple flow types that are relevant at this spatial scale, planar Poiseuille flow (Fig. 10 center) and Couette flow (Fig. 10 right). Following Richardson (2011a, 2011b) these concepts consider a Newtonian fluid, which is incompressible and bounded by parallel plates. While one of them can move relative

to the stable other in Couette flow ("drag-induced flow"), they are fixed during Poiseuille flow (Brennen, 2006; Richardson, 2011a, 2011b; Natarov and Conrad, 2012, based on Couette, 1890). Since the latter case requires a changing pressure field it is classified as "pressure-induced flow" or "channel flow" (Richardson, 2011b; Natarov and Conrad, 2012, based on Poiseuille, 1840a, 1840b, 1840c). The resulting laminar flow in each case can be considered fully developed. However, the velocity-depth distributions follow a linear slope in Couette flow, and a parabolic curve for Poiseuille flow (Brennen, 2006; Richardson, 2011a,

2011b). Each of them is associated with simple-shear deformation (see e.g. Brennen, 2006), but the velocity distribution differs per flow type; shear strain does too. Natarov and Conrad (2012) indicate that the related orientation and strength of deformation are depth-independent for Couette flow, different than for Poiseuille flow. For the latter, maximum strain occurs at layer boundaries and decreases towards the center. Poiseuille flow generates twice as much strain for the same maximum displacement (see Appendix B) and would therefore be expected to be more effective at generating seismic anisotropy. At stronger levels

of deformation saturation may occur (see e.g. Ben Ismaïl and Mainprice, 1998), and therefore deformation may not linearly





map into strength of anisotropy. Yet, Couette and Poiseuille flow are expected to produce similar fast orientations of anisotropy.

Following Natarov and Conrad (2012), the SAF model we used here in fact illustrates Couette flow, deforming the asthenosphere due to the relative movement of lithosphere and mantle convection, generating LPO (based on McKenzie, 1979; Ribe,

1989; Karato and Wu, 1993; Richards et al., 2001). However, Couette flow may be expected to control intraplate domains, except e.g. near hotspots (based on Turcotte and Schubert, 2002). On the other hand Poiseuille flow occurs in areas of changing horizontal pressure, as in the vicinity of subduction zones (Natarov and Conrad, 2012, and references therein), holding the potential to cause a trench-parallel flow (based on Phipps Morgan et al., 2007; Long and Silver, 2009). Such a flow may be occurring in the Alpine area, driven by the changing pressure field due to the Apenninic slab rollback. This would cause

channeled flow around the European slab (under the Western Alps), making a Poiseuille flow contribution seem likely, and it would explain the mountain-chain-parallel pattern of $\phi$ in the Central Alps.

Such a contribution has been recently revealed by Barruol et al. (2019) investigating a possible plume-ridge interaction, linked by a channel flow near Réunion, as proposed by Morgan (1971) and extended in Morgan (1978). Revealing a coincidence in

the SWS (based on Scholz et al., 2018) and surface-wave fast orientations (based on Mazzullo et al., 2017), this study has found a roughly $100 - 150\,\mathrm{km}$ thick asthenospheric flow narrowing eastwards toward the ridge. We estimated the depth extent (thickness $h$) of the deforming region beneath the Central Alps through Koulakovs tomography as that of the low-velocity (high-temperature) zone in profile B, which corresponds to depth intervals between $\sim 160 - 340\,\mathrm{km}$ (southwest) and $\sim 80 - 280\,\mathrm{km}$ (northeast) and thus a thickness of the deforming zone of $180$ to $200\,\mathrm{km}$. This inference may depend somewhat on the

reference velocity model, but since the Koulakov model images a much wider region, it is quite clear that the reference model (ak135 by Kennett et al., 1995; EuCRUST-07 by Tesauro et al., 2008) cannot assimilate a local anomaly in the Central Alps. The depth extend in the study of Barruol et al. (2019) is thus likely smaller than beneath the Central Alps, but still of comparable size. In that study, the depth distribution of fast orientation shows a spindle-shaped behavior, and the anisotropy strength is maximized at the layer center, while zero-crossings coincide with the structural boundaries (lithosphere/mesosphere). The

presence of a maximum regarding the expected minimum (based on Natarov and Conrad, 2012; Lin et al., 2016) reflects the difference between theoretical isoviscosity and temperature-related conditions, leading to reduced viscosity inside. Similar observations might be possible in the Alpine region, but a comparable study has not been accomplished yet. Although, it would be worthwhile, since that helped to rule out a recent Couette flow contribution near Réunion, based on the depth and azimuthal distribution of related fast orientations (Barruol et al., 2019, and references therein).

Now we return to the Alps and to the question whether a pure Couette flow, as assumed in the SAF model, can explain the azimuthal distribution of $\phi$ in the Central Alps. Figure 11 compares $\phi$ with surface motions derived from GNSS measurements by Sánchez et al. (2018), to see whether mantle and surface deformation can be related to each other. At first site geodetic motions and fast orientations seem to have little relation to each other. The reference system in that study was stable Eurasia,

and the question is whether another reference model can be found that renders geodetic motions and fast orientation similar.





Comparison reveals an opposite sense of rotation: while surface motion implies a sinistral sense of rotation with a progressively larger northward motion toward the east, such a rotation is not apparent in the fast orientations, rather the opposite (and neither in the tomographic images). If only Couette flow were active, the two orientations should be more or less similar (see e.g. Silver, 1996 and Barruol et al., 2019), as consistent large-scale surface motion should be controlled by the relative plate motion of Europe. Hence, the observed flow is likely (partly) decoupled from the lithospheric movement (see e.g. Barruol et al., 2019).

Poiseuille flow is therefore the more likely deformation model for the area. Such a flow would be consistent with the vertical alignment of olivine b-axes found in our study. Since we assume the flow in the north to be related to the pulling force of the Apenninic slab rollback, a change in the pressure field can be expected. From a geodynamic point of view, a Poiseuille contribution does not require involvement of the transition zone, which in principle could be investigated by inspecting temperatures in the transition zone, e.g. from receiver functions. However, the flow around the Alps, is perhaps too spatially confined to be resolved by the necessarily long-period receiver functions.

Couette and Poiseuille flow can occur simultaneously; this is referred to as plane Couette-Poiseuille flow (Natarov and Conrad, 2012, based on Papanastasiou et al., 2000; Lenardic et al., 2006; Höink and Lenardic, 2010; Crowley and O'Connell, 2012). Natarov and Conrad (2012) find that $40\,\%$ of the flow at global scale occurs as Poiseuille flow. For Central Europe, a similar value is obtained; the underlying convection model does perhaps not have sufficient spatial resolution though to make such inferences for our region of interest. Future studies might further illucidate the relative importance of Couette and Poiseuille flow for our area. Occuring together these flow types would produce a changing orientation of shear, if pressure in an orthogonal direction to the plate movement is not constant. The appearance of channeled flow can then influence azimuthal anisotropy derived from surface-waves, as e.g. discussed before for the study by Barruol et al. (2019).

The interpretation in the southern part, where we have observed a considerably weaker fit with each of the anisotropic models and weaker splitting, is more difficult as the region is affected by different processes. On one hand, a similar flow as for the northern part could affect the SWS, but whether the splitting occurs in the deep low-velocity region beneath the European slab detachment in profile A remains an open question considering the presence of a strong low-dVp Po plain anomaly (Lippitsch et al., 2003, based on Spakman et al., 1993; Solarino et al., 1996; Bijwaard and Spakman, 2000 or Koulakov et al., 2009) on top of the slab. If this area can be understood as hydrated (see e.g. Giacomuzzi et al., 2011, and for further reading Hearn, 1999), serpentine, known to react highly anisotropic (see e.g. Katayama et al., 2009 or Salimbeni et al., 2018, based on Bezacier et al., 2010), must be further considered. However, assuming A-type olivine a flow crossing the slab can be ruled out as origin of the observed anisotropy, as it would generate a mountain-chain-perpendicular SWS pattern (see e.g. Eakin et al., 2010). The surface motions derived by Sánchez et al. (2018) in the southern subarea (Fig. 11 bottom row) are slightly stronger (0.35 - 0.78 mm/a), and mainly oriented towards NNW. The deviations from the general NE-SW fast orientation appear even more clearly. Besides asthenospheric flow the lithospheric contribution (see e.g. Fry et al., 2010 or Salimbeni et al., 2018) is likely increased in the southern part, following the steep incidence of the slab (see e.g. Salimbeni et al., 2018 or Kästle et al., 2019).





An indication for this is provided by a zone of weaker positive vertically-integrated travel times at $11°$ longitude. This area has been suggested before to represent a discontinuity, potentially separating the Western/Central and Eastern European slabs (see e.g. Kästle et al., 2019, based on Lippitsch et al., 2003; Koulakov et al., 2009, Mitterbauer et al., 2011; Zhao et al., 2016; Hua et al., 2017). Here also the splitting decreases along profile C, suggesting that indeed slab anisotropy plays a more important role in the southern part.





## 6 Conclusions

In this study we have applied the non-vertical-ray shear-wave splitting approach to high-quality shear-wave splitting measurements of previous studies in the Central Alps to further constrain the cause of seismic anisotropy in a single-layer case. We have compared modeled and derived angular shear-wave splitting variations $\delta\phi$ and $\delta\Delta t$ and took advantage of the polarity

difference of the oscillation parameter $d_1$ for olivine in a b-up or c-up configuration to distinguish between more recent flow related deformation in the asthenosphere (Simple Asthenospheric Flow) and frozen-in deformation into lithosphere (Vertical Coherent Deformation). Initially we investigated the backazimuthal distribution in $\delta\phi$ and $\delta\Delta t$ of the whole study area at once, revealing that the roughly mountain-chain-parallel shear-wave splitting pattern, with $\Delta t$ generally varying between $\sim 0.48$ - $2.88\,\mathrm{s}$, can be related to a b-up olivine orientation. A horizontal foliation and flow plane of deformation are thus likely. How-

ever, the comparison of the shear-wave splitting pattern with a $dVp$ model demonstrated that the observed fast orientations are more likely aligned with the spatial distribution of velocity anomalies rather than being simply mountain-chain-parallel. Using the transition between slower and faster regions considering vertically-integrated travel times, we distinguished a northern and southern subarea, which revealed the following differences. While $\phi$ rotates along the Alps in the northern subarea and $\Delta t$ tends to accumulate between $\sim 1.28$ - $1.92\,\mathrm{s}$, $\phi$ shows only little rotation in the southern subarea, where $\Delta t$ is rather smaller

(0.96 - $1.76\,\mathrm{s}$). Despite the problem of a reduced backazimuthal coverage when considering subareas, the $\delta\phi$ and $\delta\Delta t$ variation still suggests a b-up olivine orientation for the northern subarea. However, the comparison of fast orientations with crustal motion revealed opposite sense of rotation, and hence little relation between mantle and current crustal deformation. This suggests that the flow in the Central Alps nowadays cannot be explained by a pure Couette flow as assumed in the Simple Asthenospheric Flow model. Instead the presence of a depth-dependent plane Couette-Poiseuille flow, channeled by the European slab

geometry beneath the Alps, compensating the mass deficit and pressure difference due to the slab rollback of the Apennines, is more likely. In the southern subarea the behavior of $\delta\phi$ and $\delta\Delta t$ is not uniquely explainable by one of the investigated end-member models so far. Besides a possible flow contribution as in the northern subarea below the European slab detachment, effects from serpentinization above the slab might possibly occur. The spatial correlation between shear-wave splitting and vertically-integrated travel times depth renders a contribution of the lithospheric slab likely. Finally, our study showed, that

even an initially simple-looking shear-wave splitting pattern might reveal unexpected complexity and lead to crucial insights.




## Appendix A: Vertically-integrated travel times through the model of Hua et al. (2017)

To test our separation into subareas based on the tomography model of Koulakov et al. (2009), we further calculate the vertically-integrated travel times also for the model of Hua et al. (2017) (reference model: iasp91 by Kennett and Engdahl, 1991; EuCRUST-07 by Tesauro et al., 2008). Similar as in Fig. 6, the $dVp$ depth slices in Fig. A1 are strongly influenced

by the occurrence of high-$dVp$ anomalies, related to the Western/Central European slab and the western end of the Eastern European slab, surrounded by a flow system (low-$dVp$ anomalies). However, different from the tomography of Koulakov et al. (2009), this pattern is less obvious at $90\,\mathrm{km}$, but with increasing depth the complexity develops. In comparison to the low-$dVp$ zone in Fig. 6, associated with the counterflow mentioned before (Salimbeni et al., 2018), the anomaly is laterally more focussed here and connected to an even stronger low-$dVp$ Po plain anomaly (Lippitsch et al., 2003, based on Spakman et al.,

1993; Solarino et al., 1996; Bijwaard and Spakman, 2000 or Koulakov et al., 2009), possibly explainable by a hydrated Adriatic mantle wedge (see e.g. Giacomuzzi et al., 2011, and for further reading Hearn, 1999). The slabs are thus separated more clearly, but a mountain-chain-perpendicular flow in between is not indicated by the fast orientations. Overall, the SWS pattern seems to correlate more with the spatial distribution of $dVp$ anomalies in the model of Koulakov et al. (2009), but as individual depth slices might not be indicative enough, we also determine the vertically-integrated travel times subsequently (Fig. A2). The

slower regions also appear around the faster areas; yet their appearance is somewhat different: the depth range differs slightly (65 - 410 km) from Fig. 7 (77.5 - 410 km); yet the reference models differ. Also the negative and positive anomalies tend are stronger in the model of Hua et al. (2017). Across the whole upper mantle the high complexity in comparison to the model of Koulakov et al. (2009) indeed remains. A differentation into a northern and southern subarea would be sligthly different following Hua et al. (2017), and more complex. For this reason we decided to use the tomography of Koulakov et al. (2009)

for the separation.





## Appendix B: Deformation from flow

Here we give some details on the two deformation models presented in Fig. 10, the planar Poiseuille model as well as the Couette model (after Brennen, 2006). Both describe steady laminar flow between two infinitely long parallel plates. The difference is that in Couette flow one of the plates has a velocity $U$ relative to the other, which is assumed to be at rest, for the sake of the
5    argument. There is no pressure gradient in the fluid. In contrast, the Poiseuille model presents the case, where both plates are at rest, and the flow is caused by a lateral pressure gradient, parallel to the plates.

Assuming that the only non-zero component of the flow is $U_x$, and that velocity and pressure are independent of time, the continuity equation for an incompressible fluid is

$$\frac{\partial U_x}{\partial x} = 0 \tag{B1}$$

and $U_x(z)$ is a function of $z$ only. For an incompressible fluid with constant viscosity $\eta$, the flow equations become

$$\frac{\partial p}{\partial x} = \eta \frac{\partial^2 U_x}{\partial z^2}\,, \tag{B2}$$

and

$$\frac{\partial p}{\partial z} = 0\,. \tag{B3}$$

15    The pressure $p$ is a function of $x$ only, and the flow velocity becomes

$$U_x = \frac{1}{\eta}\left(\frac{dp}{dx}\right)\frac{z^2}{2} + C_1 z + C_2 \tag{B4}$$

with the integration constants $C_1$ and $C_2$.

For Couette flow we have $dp/dx = 0$, and no-slip conditions at the upper and lower boundaries provide

$$U_x^{Cou} = \frac{U_{max} \cdot z}{h} \tag{B5}$$

with maximum velocity $U_{max}$ and channel thickness $h$. The velocity gradient within the channel is

$$\frac{\partial U_x^{Cou}}{\partial z} = \frac{U_{max}}{h}\,. \tag{B6}$$

Poiseuille flow with no-slip conditions yields

$$U_x^{Poi} = \frac{1}{\eta}\left(-\frac{dp}{dx}\right)\frac{z}{2}(h-z)\,. \tag{B7}$$

Assuming a constant lateral pressure gradient $-dp/dx = A$, the velocity gradient is

$$\frac{\partial U_x^{Poi}}{\partial z} = \frac{A}{\eta}\left(\frac{h}{2} - z\right)\,, \tag{B8}$$





which attains its maximum value $\frac{Ah}{2\eta}$ at the top of the channel. For convenience, we introduce a scaled flow velocity

$$\overline{U}_x = \frac{U_x}{U_{max}} \tag{B9}$$

relative to the maximum velocity $U_{max}$. For Poiseuille flow, the maximum velocity occurs in the center of the channel and it is

$$U_{max}^{Poi} = \frac{A}{\eta}\frac{h^2}{8}. \tag{B10}$$

The scaled velocity gradients are therefore

$$\frac{\partial \overline{U}_x^{Cou}}{\partial z} = \frac{1}{h} \tag{B11}$$

and

$$\frac{\partial \overline{U}_x^{Poi}}{\partial z} = \frac{8}{h^2}\left(\frac{h}{2} - z\right). \tag{B12}$$

The absolute level of strain rate (or velocity gradient) $|\partial \overline{U}_x/\partial z|$ determines the increase in seismic anisotropy at a given depth and time. For long periods (Montagner et al., 2000; Wüstefeld et al., 2009), SKS splitting can be related to its vertical integral

$$\int_0^h |\frac{\partial \overline{U}_x}{\partial z}|dz. \tag{B13}$$

This assumes that no saturation effects occur in the deformation-anisotropy relation (see e.g. Ben Ismaïl and Mainprice, 1998). It follows that Poiseuille flow is twice as efficient in generating strain (and SKS splitting) compared with Couette flow, for a

given maximum velocity (which can be constrained from geodynamics/tectonics).



*Data availability.* Shear-wave splitting measurements of Barruol et al. (2011), Qorbani et al. (2015), and Salimbeni et al. (2018) are publicly available. Koulakov et al. (2009) and Hua et al. (2017) provided their $dVp$ models. We have used topographic maps and coordinates of deformation front data from Amante and Eakins (2009), Ferranti and Hormann (2014) and 4D-MB SPP (2019), and the map of horizontal deformations from Sánchez et al. (2018).

5 *Author contributions.* The work presented here was carried out and written down by the first author. Appendix B was accomplished by the co-author, who also controlled texts and figures. Related suggestions were included by the first author.

*Competing interests.* We declare that no competing interests are present.

*Acknowledgements.* We thank Irene Bianchi for her comments and suggestions, which helped to improve the section on tectonic settings in the Alps. Further we want to acknowledge all data suppliers mentioned above, and the operators of the Swiss Digital Seismic Net-
10 work/Switzerland Seismological Network (10.12686/SED/NETWORKS/CH), the Austrian Seismic Network (10.7914/SN/OE) and Italian National Seismic Network (10.13127/SD/X0FXNH7QFY) for maintaining seismological stations and providing data, leading to the shear-wave-splitting datasets studied here. Part of this work was supported by a Dissertation Completion Fellowship of the University of Vienna, and the Austrian Science Fund FWF through projects numbers P26391 and P30707.





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

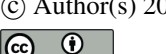



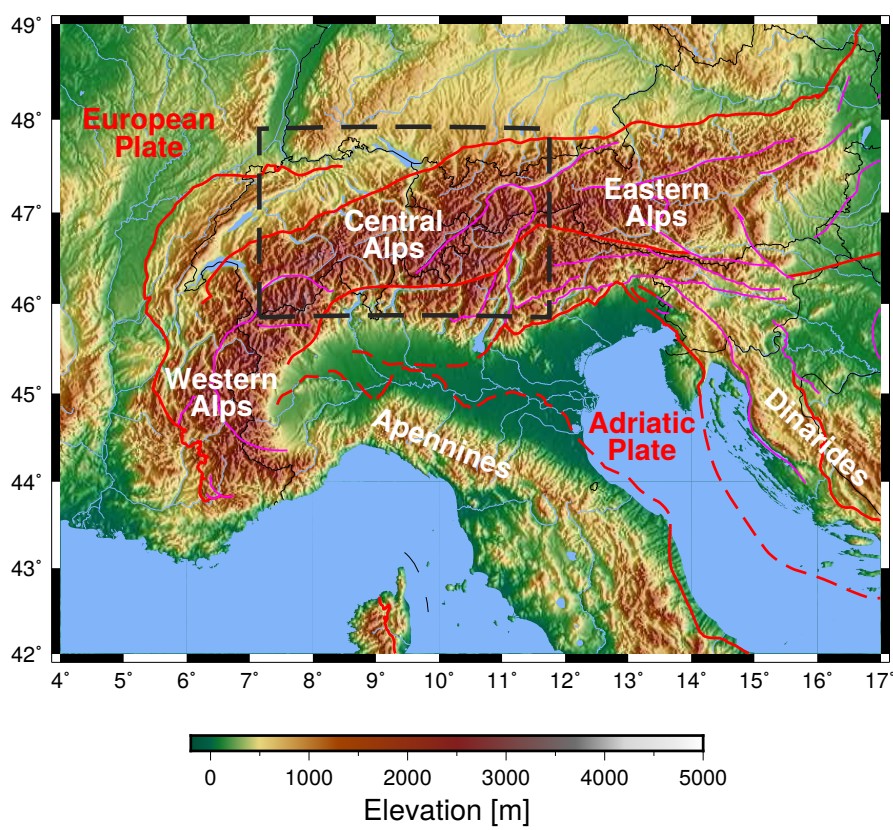

**Figure 1.** The study region in the Central Alps (dashed rectangle) on a topographic map (Amante and Eakins, 2009). Deformation fronts (red solid: exposed, red dashed: subsurface, magenta: Neogene fault) are taken from the Alpine geological map (4D-MB SPP, 2019, based on Schmid et al., 2004; Schmid et al., 2008).





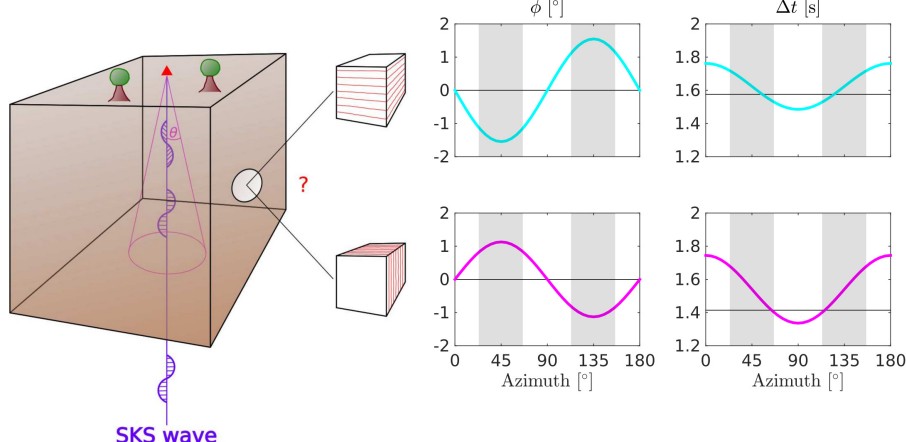

**Figure 2.** Principle of non-vertical-ray shear-wave splitting as shown in Löberich and Bokelmann (2020). (left) Geometry of SKS wave paths, along a cone through the upper mantle under a seismological station. (center) We consider the two endmember models of geodynamic interest, the Simple Asthenospheric Flow Model (above), associated with a horizontal orientation of foliation planes, and the Vertical Coherent Deformation (below), associated with a steep orientation of foliation planes. (right) Expected azimuthal variations of shear-wave splitting parameters (fast orientation $\phi$ and delay time $\Delta t$) for the two models compared with vertical incidence (black lines) assuming a mantle consisting of pure olivine [70 % b-up (top, cyan lines) or c-up (bottom, magenta lines) and 30 % isotropic]. Note that the azimuthal variation of $\phi$ is rather different i.p. around the extrema (gray-shaded intervals).





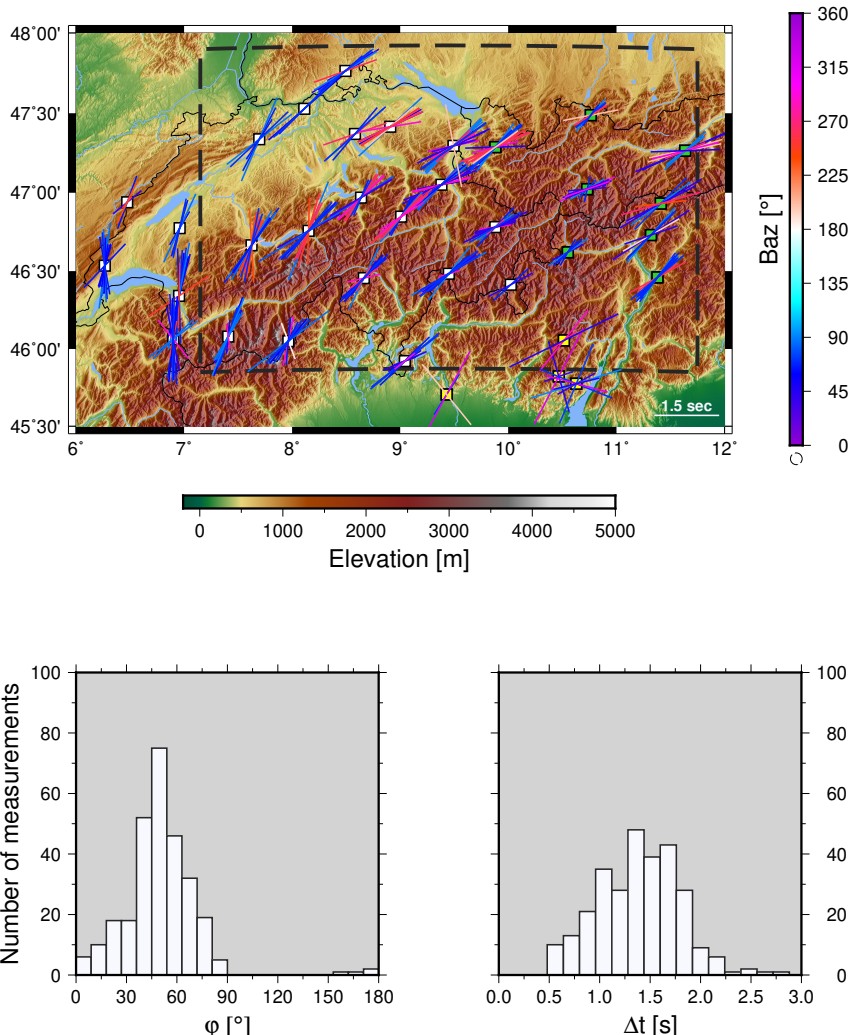

**Figure 3.** (top) High-quality ($\sigma_\phi \leq 20°$) shear-wave splitting measurements (permanent stations in white: Barruol et al., 2011; green: Qorbani et al., 2015; yellow: Salimbeni et al., 2018) in the Central Alps on a topographic map (based on Ferranti and Hormann, 2014). Line lengths give splitting delay $\Delta t$, orientation the fast orientation $\phi$. The dashed rectangle surrounds the study area as in Fig. 1. (bottom) Histograms of measurements in the study area. Fast orientations (bottom, left) indicate a unimodal distribution around NE-SW related to the clockwise rotation of $\phi$. Delay times (bottom, right) occur in a range between $\sim 0.48 - 2.88\,\mathrm{s}$.





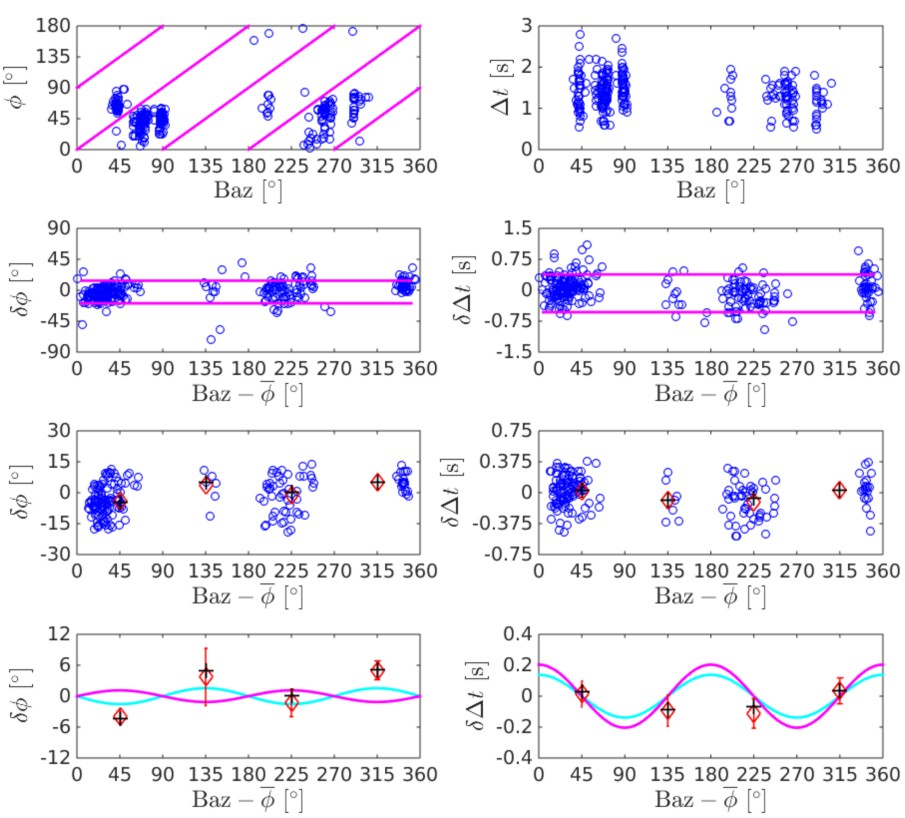

**Figure 4.** Summary of the shear-wave splitting dataset: (top) High-quality measurements of fast orientations $\phi$ (left) (solid magenta lines give Null orientations) and delay times $\Delta t$ (right). (second row) Variations $\delta\phi$ and $\delta\Delta t$ from station averages, shown for backazimuths relative to the station averages. Thresholds for $\delta\phi$ and $\delta\Delta t$ (solid magenta lines) are related to the distribution. (third row) Means and medians (diamonds and pluses) of values from within thresholds, and angular intervals of $\pm 33.25°$ adapted from Fig. 2. (bottom) Comparison of derived means (related errorbars show the $2\sigma$-error) and medians with expected variations from Fig. 2 for a b-up (cyan) and c-up (magenta) olivine orientation.





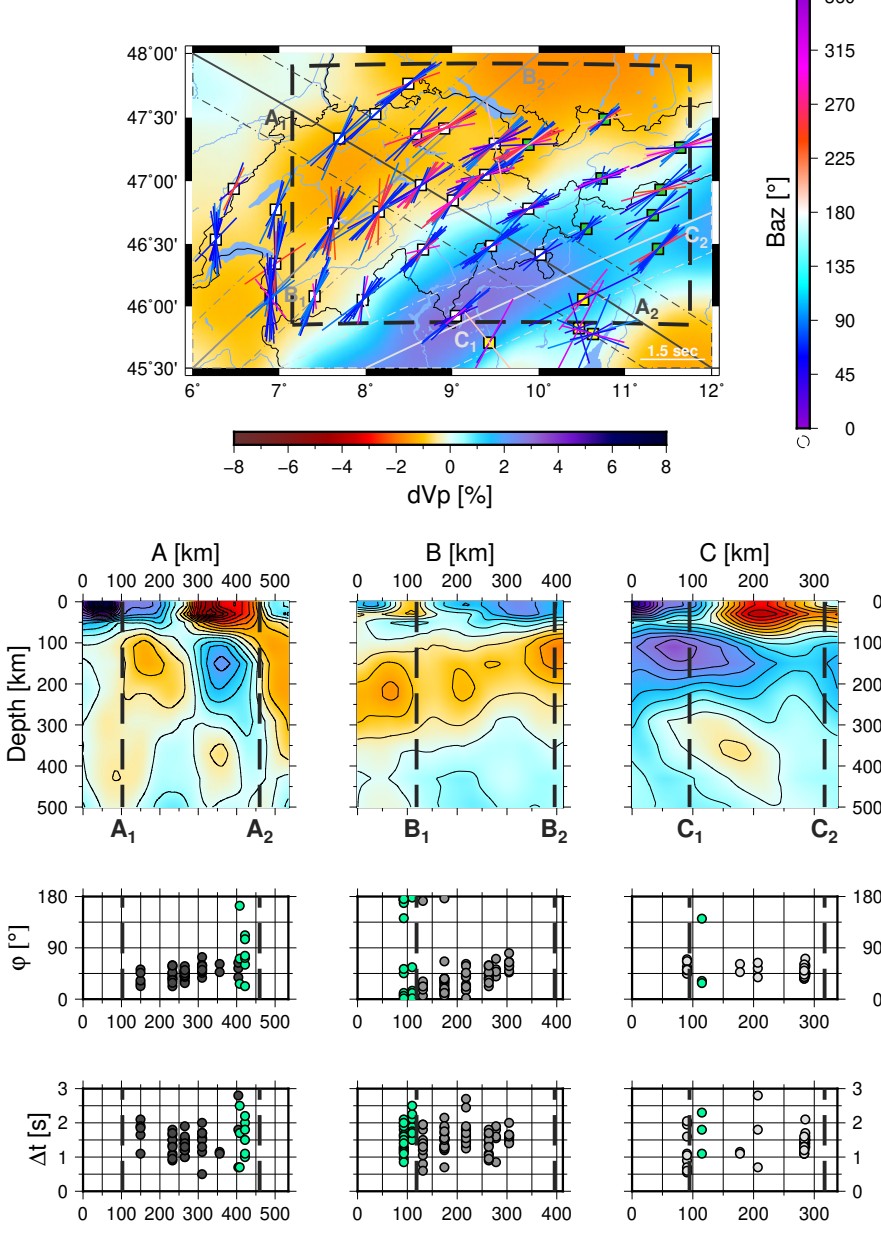

**Figure 5.** Comparison of the shear-wave splitting measurements and P-wave velocity perturbations in the model of Koulakov et al. (2009). (top) Spatial shear-wave splitting pattern on the $dVp$ slice in 150 km depth. Slower regions in the study area are associated with hotter and thus more deformable material, while faster regions are related to colder lithospheric structures as e.g. a slab. The $dVp$ depth distribution (second row) along a NW-SE (A), SW-NE (B) and S-E (C) profile (top; gray-shaded lines; boundaries dashed) and the related variation of the fast orientation $\phi$ (third row) and delay time $\Delta t$ (bottom) are shown below. Measurements projected from outside the study area (mint-colored circles) are added for completeness.


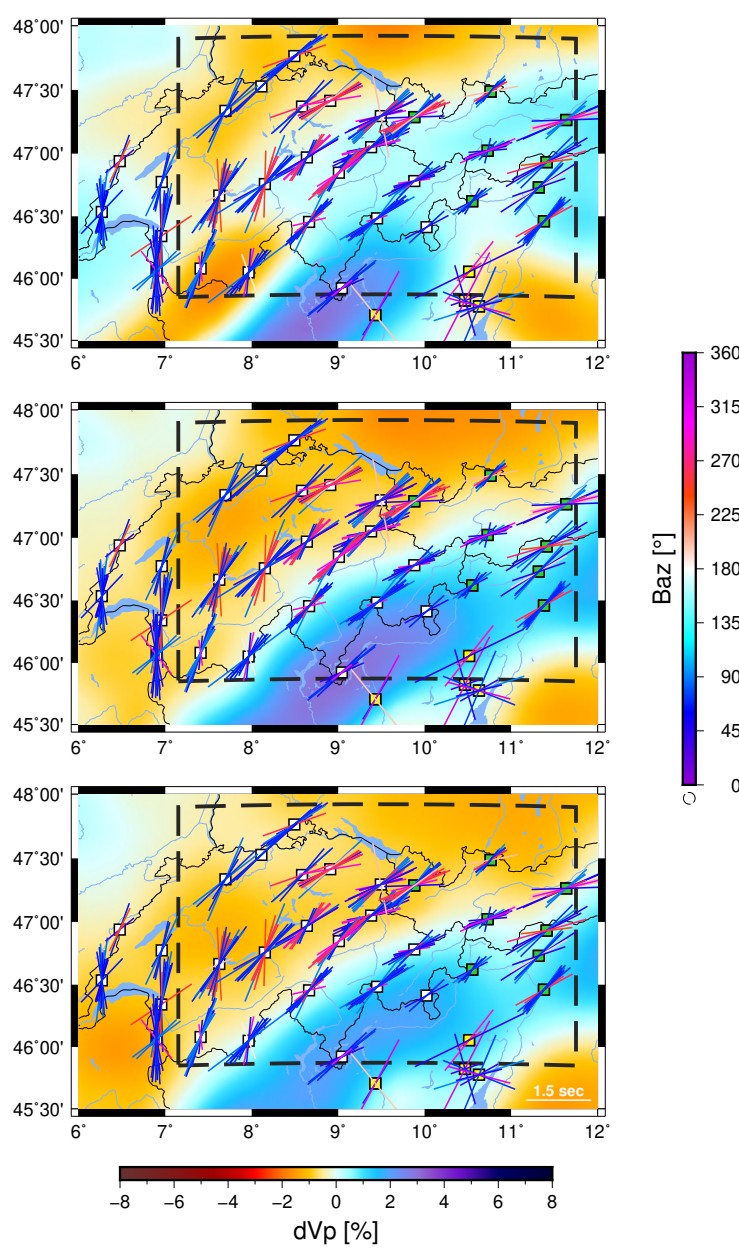

**Figure 6.** Spatial correlation of the shear-wave splitting pattern and the $dVp$ model of Koulakov et al. (2009) at depths of 90 km (top), 140 km (middle) and 190 km (bottom) in the upper mantle.



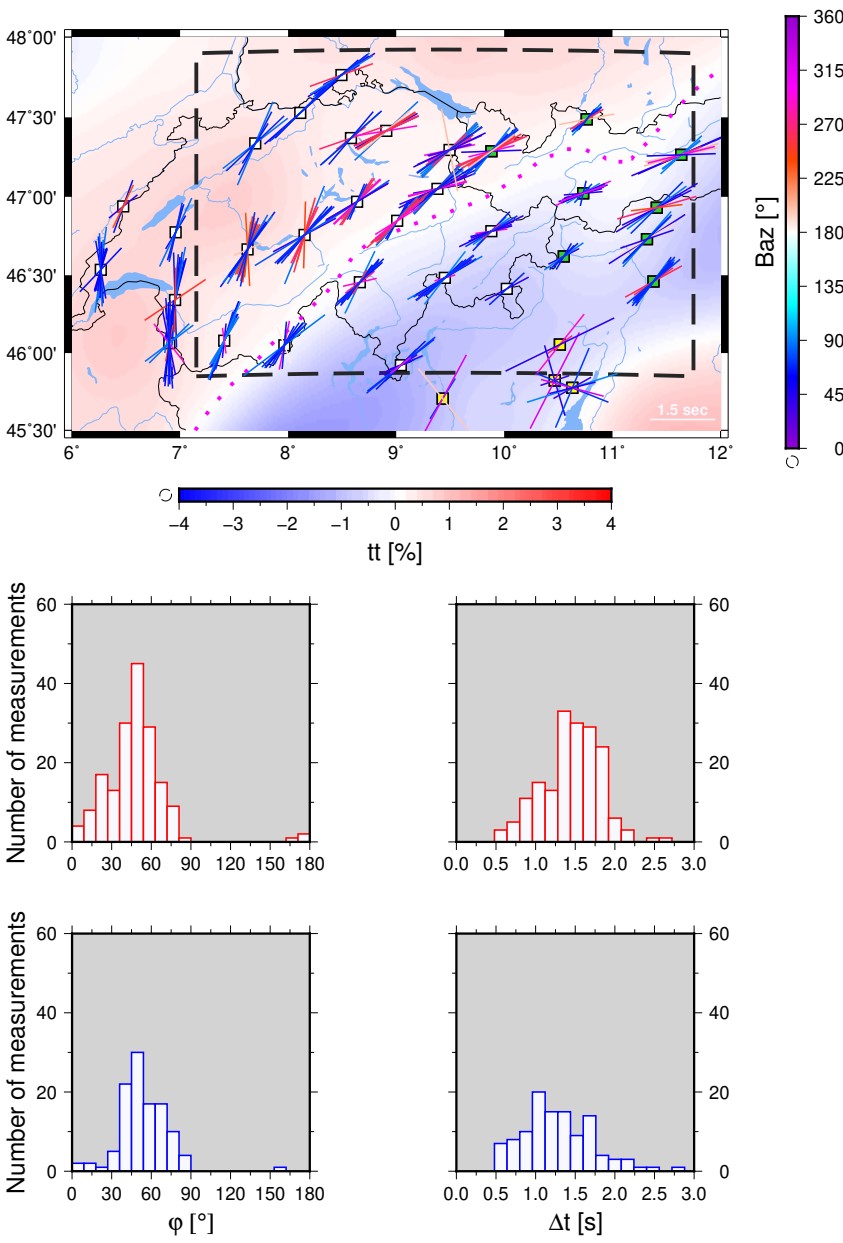

**Figure 7.** (top) Spatial correlation of the shear-wave splitting pattern with vertically-integrated travel times $tt$ through the $dVp$ model of Koulakov et al. (2009), from $77.5 - 410$km depth. Faster times are related to the presence of the slab, while slower times are more influenced by asthenospheric material. The dashed magenta line outlines the boundary between perturbations, separating a northern and southern subarea. Comparison of histograms of fast orientations $\phi$ (bottom, left) and delay times $\Delta t$ (bottom, right) reveal a noticeable larger rotation of $\phi$ and a tendency to higher $\Delta t$ in the northern subarea (second row) than in the southern subarea (bottom).



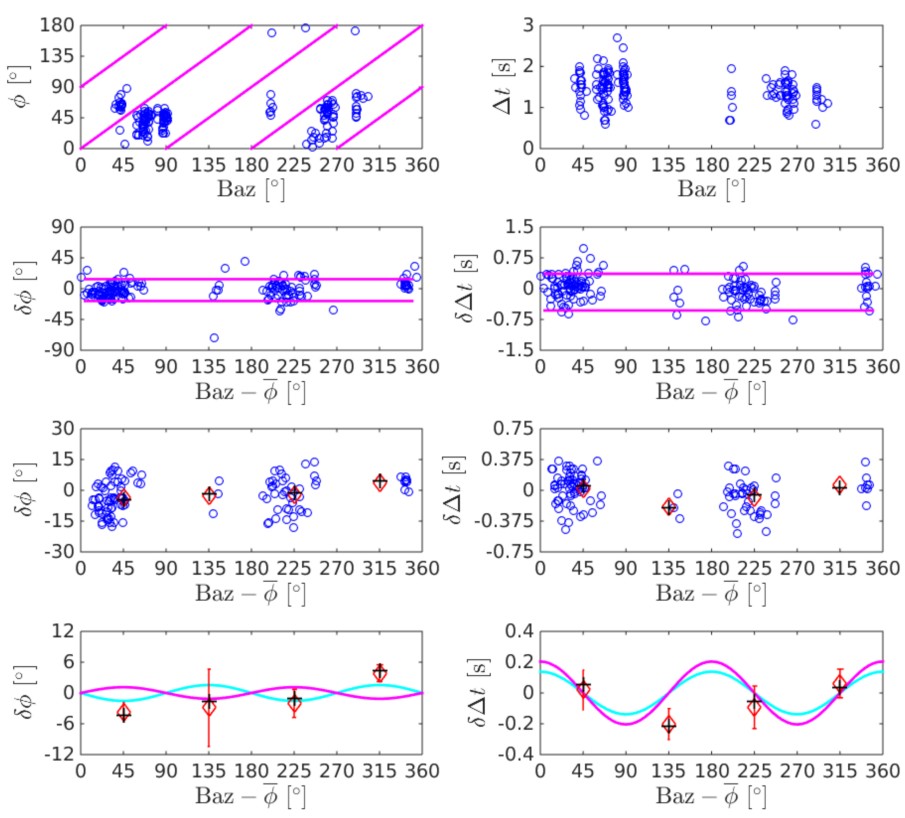

**Figure 8.** Summary of the northern subarea. (top) High-quality measurements of fast orientations $\phi$ (left) (solid magenta lines give Null orientations) and delay times $\Delta t$ (right). (second row) Variations $\delta\phi$ and $\delta\Delta t$ from station averages, shown for backazimuths relative to the station averages. Thresholds for $\delta\phi$ and $\delta\Delta t$ (solid magenta lines) are related to the distribution. (third row) Means and medians (diamonds and pluses) of values from within thresholds, and angular intervals of $\pm33.25°$ adapted from Fig. 2. (bottom) Comparison of derived means (related errorbars show the $2\sigma$-error) and medians with expected variations from Fig. 2 for a b-up (cyan) and c-up (magenta) olivine orientation.



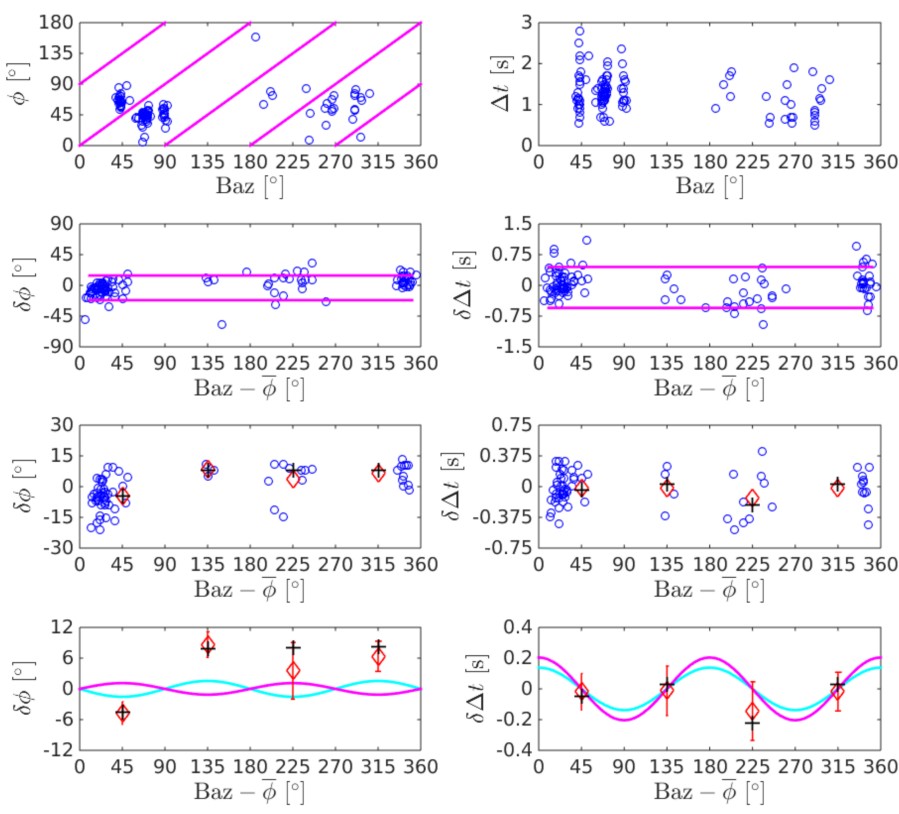

**Figure 9.** Summary of the southern subarea. (top) High-quality measurements of fast orientations $\phi$ (left) (solid magenta lines give Null orientations) and delay times $\Delta t$ (right). (second row) Variations $\delta\phi$ and $\delta\Delta t$ from station averages, shown for backazimuths relative to the station averages. Thresholds for $\delta\phi$ and $\delta\Delta t$ (solid magenta lines) are related to the distribution. (third row) Means and medians (diamonds and pluses) of values from within thresholds, and angular intervals of $\pm 33.25°$ adapted from Fig. 2. (bottom) Comparison of derived means (related errorbars show the $2\sigma$-error) and medians with expected variations from Fig. 2 for a b-up (cyan) and c-up (magenta) olivine orientation.





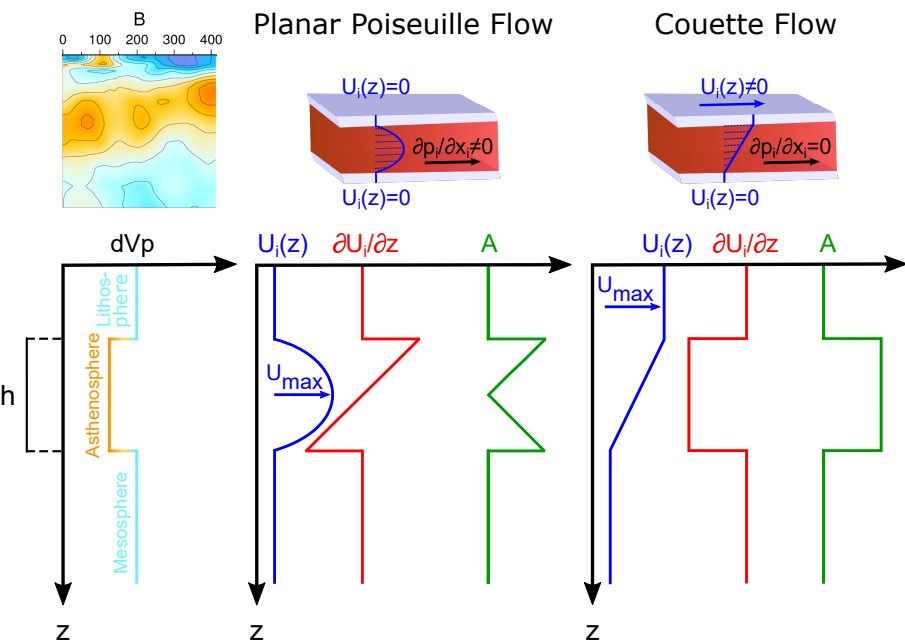

**Figure 10.** The two deformation models considered for the upper mantle; (left) Seismic low-velocity zone indicating the deforming zone, (center) planar Poiseuille flow (channel flow), and (right) Couette flow (see text and Appendix B). Blue colors indicate flow velocity, red colors the vertical gradient of flow, and green the related deformation (based on Brennen, 2006; Richardson, 2011a, 2011b), and anisotropy (Barruol et al., 2019).



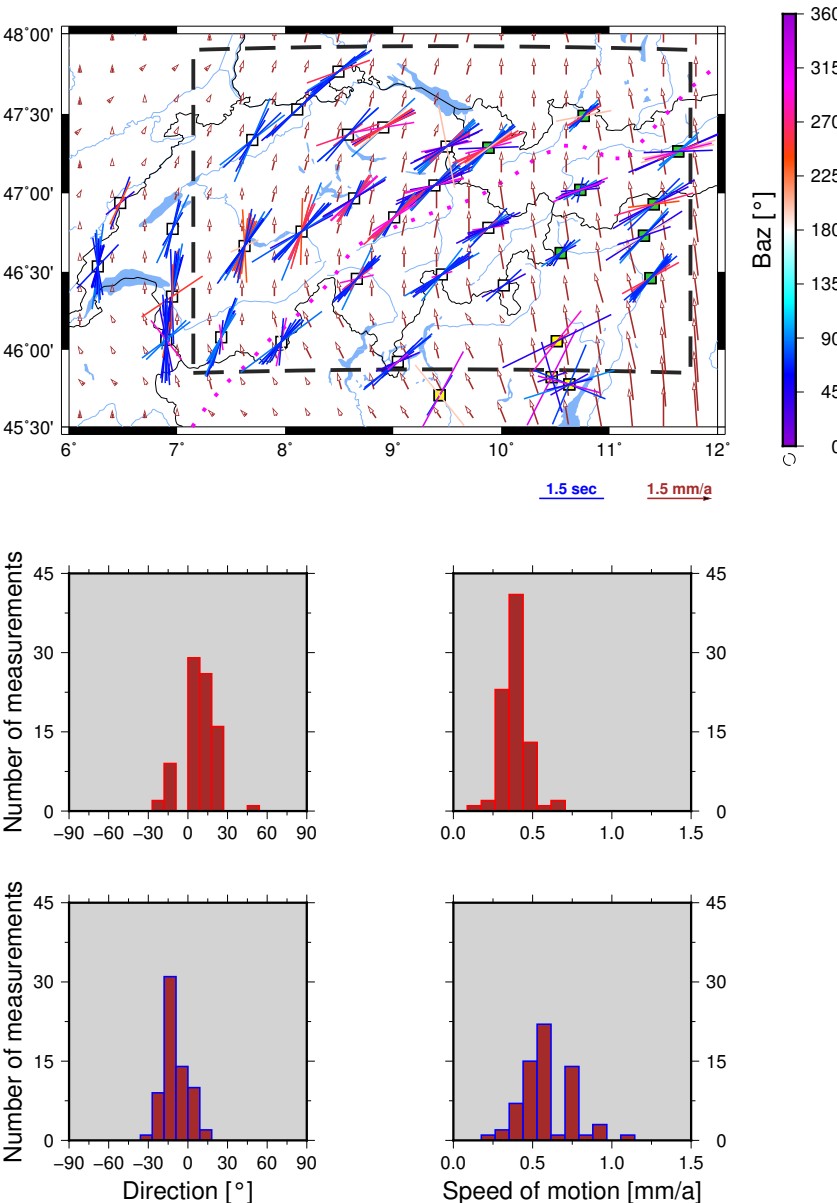

**Figure 11.** (top) Comparison of the shear-wave splitting measurements with surface motions (brown arrows) interpolated from GNSS data by Sánchez et al. (2018). The crustal motion pattern generally differs from the fast orientations $\phi$ (see text). Histograms of the direction (left) and the speed of motion (right) reveal a tendency to small, NNE oriented motions in the northern subarea (second row) and slightly stronger, NNW oriented motions in the southern subarea (bottom).


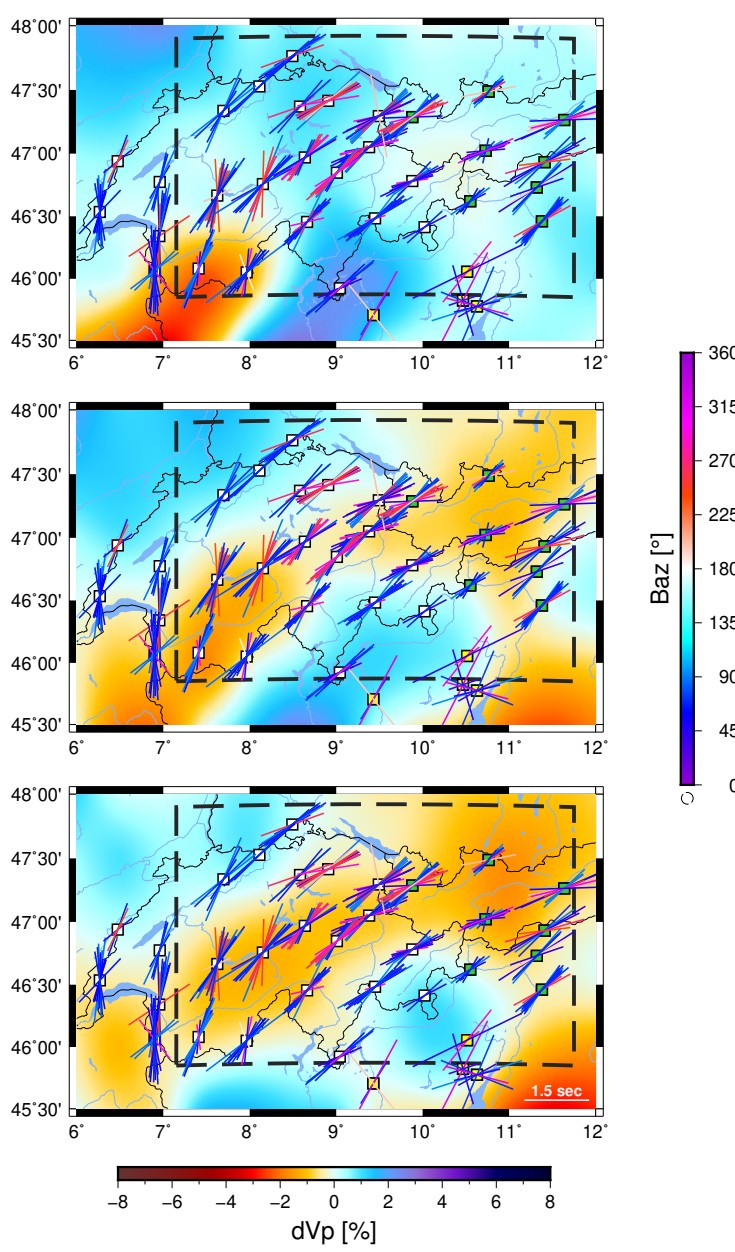

**Figure A1.** Spatial correlation of the shear-wave splitting pattern and the $dVp$ model of Hua et al. (2017) at depths of 90 km (top), 140 km (middle) and 190 km (bottom) in the upper mantle.


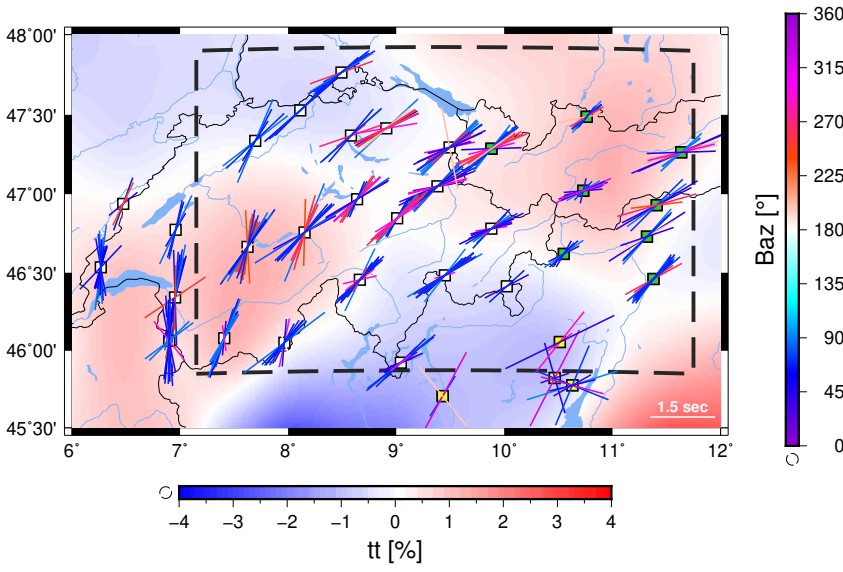

**Figure A2.** Spatial correlation of the shear-wave splitting pattern with vertically-integrated travel times $tt$ through the $dVp$ model of Hua et al. (2017), from $65 - 440\,\mathrm{km}$ depth. Faster times are related to the presence of the slab, while slower times are more influenced by asthenospheric material.