# Peer review of "Mantle flow under the Central Alps: Constraints from non-vertical SKS shear-wave splitting"

_Solid Earth, 2020_

## Referee Comment (RC1) · Anonymous Referee #1 · 1 Apr 2020

In this article, Löberich and Bokelmann present a new study on the cause of seismic anisotropy in the upper mantle beneath the Central Alps. After selecting high-quality, pre-existing shear-wave splitting measurements from several seismic networks, the authors apply a recently developed technique that describes the azimuthal dependence of shear-wave splitting parameters in the case of non-vertical rays of core-refracted phases. The authors integrate the resulting shear-wave splitting measurements with insights from tomographic velocity models and surface deformation directions, concluding that a Couette-Poiseuille flow is likely responsible for the fast-polarization orientations observed.

[Figure]

I recognise there is some effort in putting the paper together, however, the article is poorly presented. English is below standard for publication with outright errors of spelling, grammar and faulty construction of sentences that very often compromise the flow and understanding of the article. Similarly, the structure of the paper is far from being acceptable (see comments below) and figures are frequently not thoroughly described. Not being a native-English speaker myself, I understand the struggle and frustration of writing in a foreign language, but I always make sure that a native speaker reads my drafts. This is the main advice I would give the authors.

While I believe that there can be value in this study, written English along with lack of structure and logic often prevent the reader from understanding what the authors want to convey. I am afraid that this must be revised before the manuscript can be accepted for publication, or even go through a more technical review. I recommend major revision of the paper and highlight the need for this to be implemented. In principle, the revision effort should not be overwhelming if the authors have access to a native speaker.

Below are some recommendations that could improve the structure of the manuscript. I am not providing any corrections about the syntax and grammar as this would be too onerous:

- As a general comment, I find the paper to be unnecessary long. Key messages are too often buried in text. Answers to the questions "Why did you do this work?", "What did you do?" and "What did you achieve?" are hard to dig out. Indeed, I struggled to understand most of the paper.

- Tectonic setting is confusing and not sufficient, it needs improvement.

- Whole page 5 is clearly part of the Introduction but it is in the tectonic setting? You are discussing previous studies and stating hypothesis and objectives.

- You are mixing "Results" with "Method and Data" and "Discussion". There are parts of Section 4 (Results) that should be in Section 3 (Method and Data), and parts of

Section 4 that should be in Section 5 (Discussion). This is where the manuscript loses all its logic and flow. You should essentially reorganise all the sections but Abstract and Conclusions.

- Once all text is revised and clarity improved, it will be possible to evaluate the results of this study. As of now, speculation seems to dominate your arguments, rather than evidence-based conclusions. Also, interpretation has a large emphasis on previous studies rather than concentrating on what you are bringing to the table.

---

## Referee Comment (RC2) · Anonymous Referee #2 · 3 Apr 2020

This manuscript uses the non-vertical propagation of SK(K)S phases along with previous shear wave splitting observations to model anisotropy beneath the Alps. The authors find that beneath the northern portion of their study region, anisotropy can best be modeled with a b-up olivine alignment indicative of flow in the asthenosphere. In the southern half of the study area, the slab seems to be in the way complicating the observations and producing less well-fit model results.

Comments in no particular order:

1) Are there any restrictions placed on the delay time errors used in the high quality splitting dataset like the 20deg restriction placed on fast direction? a. What are average

errors on dts and Phis of the original splitting dataset? and the restricted dataset?

2) The manuscript would benefit from an expanded discussion of the 1-layer of anisotropy assumption. a. I think a supplemental figure of baz for the stations before restricting the dataset would help. It is important to make sure that by restricting the dataset you are not removing some of the BAZ variability. b. There seems to be large variances in dt, which can be indicative of layered splitting

3) Why are there fewer BAZs represented at stations above the slab (e.g. figure 5)? Slab stations seem to have fewer 225-360deg baz splits. If this is due to dataset error restriction, it could be an indicator of layers of anisotropy. Layered splitting measurements often have larger errors at BAZs where the largest variations in phi and dt occur.

4) How are the BAZs of 45 and 135 around which things are stacked chosen? Is it based on average fast directions? Or because of BAZ coverage? Or some initial assumption about olivine axes?

5) Figure 5 – change the colors of the cross section lines. It is very hard to see B-B'

6) Does the 70% of fully aligned olivine assumption make sense given the observed delay times? Wouldn't a ~100km layer with 70% alignment yield far larger delay times than the 1-2s dts that are measured?

7) It would be helpful to add a discussion of how b-up and c-up olivine relate (or translate) to the more traditionally used A-, B-, C-, D-, E- type fabrics. a. A-, C- and E-type fabrics have all been proposed for the asthenospheric mantle and all have general properties where phi aligns with strain. But A-type and C-and E-types have different b-up and c-up relationships to shear strain. So how do you distinguish between them, or do you have to assume A-type?

8) Figs 8 and 9 and text – It would be useful to discuss and show a null test of the models. It looks to me like a flat line model would match the data just as well as the varying BAZ model.

[Figure]

Technical: P1L1 and throughout – "e.g." is used throughout the text in cases where it is not needed. P1L2 – "constraint" to "constrain" P4L18 – "till" to "until" P14L33 – "first site" to "first sight"
* * *

---

## Referee Comment (RC3) · Anonymous Referee #3 · 28 Apr 2020

The authors aim to characterize geodynamic processes in the Western and Central Alps from the azimuthal variation of shear-wave splitting measurements of core-mantle converted S-phases (S(K)KS) considering their non-vertical incidence. The applied method allows to differentiate between a mainly lithospheric or asthenospheric origin for the measured anisotropy, as the polarity of the fast axis variation shows opposite sign. The mechanisms relate to b-up and c-up olivine alignment resulting in horizontal foliation for the asthenospheric flow model and vertical foliation for vertical coherent deformation. They apply the analysis to previously published shear-wave splitting measurements at permanent stations in the Western and Central Alps first for the full study area and subsequently to a northern and a southern subarea based on a travel time in-

tegrated dvp-model of Koulakov et al. (2009). Their findings favour the asthenospheric flow model as origin for the anisotropy for the data set of the full data set and the northern subarea. The southern subarea allows no precise statement, most likely due to the limited event coverage. They conclude, that the anisotropy in the region is originated by a Poiseuille flow driven by the Apenninic slab rollback, likely to coincide with a Couette flow linked to the absolute plate motion. The authors argue also, that a mainly lithospheric origin for the southern subregion is to be expected, due to a dominant effect of the slab.

GENERAL COMMENTS

The method applied in this paper has the potential to improve our understanding of the link between geodynamic processes and measured anisotropy. It is promising with respect to the huge data set of the previous publication, that similar results could be found here for the Central Alps indicating an asthenospheric origin.

Nevertheless, there are some major and minor issues, which give rise to questions. The study neglects important aspects and possibly equivalent models, that could explain the analysed data. Due to this incompleteness the conclusions drawn in this paper remain ambiguous, while a short discussion or comparison could likely clarify these open questions.

Generally, the submitted manuscript contains some weaknesses in language and structure. While it should be revised for grammar and wording, I would also recommend reducing the section about the tectonic setting. A more focused introduction leading to the main open questions discussed in this paper would be sufficient. The paper might benefit generally from a clearer structure.

SPECIFIC COMMENTS

As the method is based on an azimuthal variation of the splitting parameters, a more detailed discussion should be included, clarifying, why a multiple layered anisotropy

can be neglected as cause for the observed azimuthal distribution. It is not sufficient to base the single layer assumption on the spatially coherent anisotropy (page 1/line 10-11, page 3/line 8). Multiple layers with individually spatial coherent anisotropy will also produce spatially coherent distribution for the measured splitting (of course with azimuthal variation at each individual station). This question remains with respect to the limited azimuthal coverage, which appears sparser in figure 4 then described in the paper (page 8/line 5). A 90° periodicity, as expected for layered anisotropy, cannot be excluded. Therefore, I would suggest showing a fit of a two-layer anisotropy at exemplary stations to allow a meaningful discussion.

The favoured Couette-Poiseuille flow in abstract and conclusion seems contradicting to the main assumption of a single layer anisotropy resulting in an azimuthal variation of splitting parameters solely caused by the non-vertical incidence of S(K)KS-phases. A Couette-Poiseuille flow would produce a depth dependent fast axis direction resulting in azimuthal variation of phi and dt (with 90° periodicity). I agree with the first conclusion made in this paper, that the data is best explained with a Poiseuille flow model.

In the following I will list some minor suggestions for improvement

- Further topics to be discussed or mentioned -

Page 2/Line 2: LPO of olivine is not the only origin for anisotropy (Savage 1999).

Page 2/Line 6: What is about other depth regions and their contribution to anisotropy?

Page 2/Line 18-19: The elliptical motion is also depending on the frequency content of the core-mantle converted phase (see Rümpker & Silver 1998)

Page 2/Line 24: This is not generally true, as there are also other effects producing anisotropy e.g. alignment of cracks in the crust or alternating layers of different seismic velocities. (Savage 1999)

Page 3/Line 11: It would be helpful at this point to state the contradicting arguments, and also pick up on them again in the discussions section to show how the understanding of the geodynamic processes is improved by this paper (and what arguments are ruled out by this paper).

Page 6/Line 1: This section summarizes the theory found in Löberich & Bokelmann 2020. Nevertheless, it would be important to also point out the limitations and the assumptions the theory is based on (e.g. the symmetry system).

Page 8/Line 8: Does the number of measurements at one station have a large impact on the results or the observables dphi and ddt?

Page 10/Line 26-29: The southern subarea is also characterized by less data coverage. Might that also be a reason for the different observation compared to the northern subarea?

Page 13/ Line 12-13: That the data supports a high-temperature mechanism is an important conclusion here, as it is stated. While it is mentioned, that geological observations favour a different mechanism, are there geological observations in the area, that support this mechanism?

- Suggestions for reduction -

Page 2/Line 12-13: the comparison with optical anisotropy seems unnecessary at this point, as it is not used further to explain the properties of seismic anisotropy.

Page 4/Line 1: This section might be too long and detailed, as this paper aims solely to differentiate between an asthenospheric and lithospheric origin for the measured anisotropy.

Page 12/Line 12: This statement seems not to fit to the context of the publication.

Page 12/Line 23-24: It is not clear how an extended period of the AlpArray experiment is connected to the current paper.

Page 13/Line 1-6: The data set has been strongly simplified by taking the mean (and median) for the intervals around the extrema in dphi, which also coincide with ddt

expected to be zero. With this simplification it is only natural not to see any further complexities, necessary to be fitted by a more complex model. With this I don't think it is necessary to point out what complexities are not considered for the modeling, as it gives no additional information for the conclusion of this paper.

TECHNICAL CORRECTIONS

Page 2/Line 1: More generally the waves are affected by the medium they propagate through (not necessarily layers.

Page2/Line 8: effect=affect

Page 2/Line 18: "show up" = appears

Page 2/Line 19: isotropic Earth = isotropic medium

Page 2/Line 20: "shows a signal" – The measured time-series on the transverse component is not an independent signal. I would rather refer to the energy of the signal that occurs on the transverse component due to the splitting.

Page 2/Line 26: "So" seems not the right word here, as resolving the foliation seems not to be related to the weak depth resolution.

Page 4/Line 14: "e.g." seems to be unnecessary here

Page 4/Line 18: "till" = until

Page 6/Line 4: "SKS" and SKKS or S(K)KS

Page 7/Line 5: "individuals"=individual

Page 8/Line 9-10: This does not seem to be a "correction". It would be better to simply describe what is done: First the mean values are subtracted from the individual values to obtain ddt and dphi. Subsequently the backazimuth is reduced by the mean of the fast axis to shift all measurements to the same reference.

Page 8/Line 25: A "verification" might not be possible, but it might proof it as a most

probable model, which is already very promising.

Page 10/Line 21-24: This description is slightly confusing. The statements regarding the fast axis directions don't fit in the general structure of the sentence.

Page 10/ Line 26: "transition" might be the wrong word to describe a drop of magnitude in a histogram.

Page 13/Line 31: "shear strain does too"

Page 14/Line 31: "Now we return"

Page 14/Line 35+ Page 17/Line 24: "render" – seems to be the wrong word?

Page 17/Line 4: derived=measured; angular=azimuthal

---

## Author Comment (AC1) · 27 May 2020

**Comments of Referee #1**

In this article, Löberich and Bokelmann present a new study on the cause of seismic anisotropy in the upper mantle beneath the Central Alps. After selecting high-quality, pre-existing shear-wave splitting measurements from several seismic networks, the authors apply a recently developed technique that describes the azimuthal dependence of shear-wave splitting parameters in the case of non-vertical rays of core-refracted phases. The authors integrate the resulting shear-wave splitting measurements with insights from tomographic velocity models and surface deformation directions, concluding that a Couette-Poiseuille flow is likely responsible for the fast-polarization orientations observed.

1) I recognise there is some effort in putting the paper together, however, the article is poorly presented. English is below standard for publication with outright errors of spelling, grammar and faulty construction of sentences that very often compromise the flow and understanding of the article. Similarly, the structure of the paper is far from being acceptable (see comments below) and figures are frequently not thoroughly described. Not being a native-English speaker myself, I understand the struggle and frustration of writing in a foreign language, but I always make sure that a native speaker reads my drafts. This is the main advice I would give the authors. While I believe that there can be value in this study, written English along with lack of structure and logic often prevent the reader from understanding what the authors want to convey. I am afraid that this must be revised before the manuscript can be accepted for publication, or even go through a more technical review. I recommend major revision of the paper and highlight the need for this to be implemented. In principle, the revision effort should not be overwhelming if the authors have access to a native speaker. Below are some recommendations that could improve the structure of the manuscript. I am not providing any corrections about the syntax and grammar as this would be too onerous:

> *We tried to improve the structure and grammar of the manuscript during the revision and hope it is now easier to understand.*

2) As a general comment, I find the paper to be unnecessary long. Key messages are too often buried in text. Answers to the questions "Why did you do this work?", "What did you do?" and "What did you achieve?" are hard to dig out. Indeed, I struggled to understand most of the paper.

> *During the restructuring we tried to shorten the manuscript.*

3) Tectonic setting is confusing and not sufficient, it needs improvement.

> *We understand that this section might be difficult to understand, but as our paper does not focus on alpine history we see this paragraph more as background information and want to keep it short, as also Referee #3 suggests. We thus only kept page 4 and tried to improve readability.*

4) Whole page 5 is clearly part of the Introduction but it is in the tectonic setting? You are discussing previous studies and stating hypothesis and objectives.

> *We agree and shifted it to the Introduction.*

5) You are mixing "Results" with "Method and Data" and "Discussion". There are parts of Section 4 (Results) that should be in Section 3 (Method and Data), and parts of Section 4 that should be in Section 5 (Discussion). This is where the manuscript loses all its logic and flow. You should essentially reorganise all the sections but Abstract and Conclusions.

> ➢ *As the method itself is very recent and we still test their potential, it is sometimes difficult to clearly separate those section from each other. Yet we agree to some degree and shifted parts of the Result section to Method and Data. The comparison with tomography is usually part of the Discussion, but it motivates the subarea investigation and thus further results. However, we shortened the comparison with tomography somewhat and tried to avoid extensive interpretations.*

6) Once all text is revised and clarity improved, it will be possible to evaluate the results of this study. As of now, speculation seems to dominate your arguments, rather than evidence-based conclusions. Also, interpretation has a large emphasis on previous studies rather than concentrating on what you are bringing to the table.

> ➢ *We present a method that has the potential to distinguish between different deformation mechanisms and is thus able to differentiate between an asthenospheric and lithospheric cause of seismic anisotropy from shear-wave splitting measurements. Over decades this has been assumed to be impossible and the origin of observed azimuthal anisotropy patterns thus remained in question. We pick up previous models of Barruol et al. (2011) or Salimbeni et al. (2018) in the light of our new observational constraints, and expand this to types of observations (tomography, GNSS measurements, …). In agreement with previous findings our procedure reveals an asthenospheric cause (high-temperature mechanism) of anisotropy in the northern Central Alps and further constrains a Poiseuille flow type.*

---

## Author Comment (AC2) · 27 May 2020

**Comments of Referee #3**

The authors aim to characterize geodynamic processes in the Western and Central Alps from the azimuthal variation of shear-wave splitting measurements of core-mantle converted S-phases (S(K)KS) considering their non-vertical incidence. The applied method allows to differentiate between a mainly lithospheric or asthenospheric origin for the measured anisotropy, as the polarity of the fast axis variation shows opposite sign. The mechanisms relate to b-up and c-up olivine alignment resulting in horizontal foliation for the asthenospheric flow model and vertical foliation for vertical coherent deformation. They apply the analysis to previously published shear-wave splitting measurements at permanent stations in the Western and Central Alps first for the full study area and subsequently to a northern and a southern subarea based on a travel time integrated dvp-model of Koulakov et al. (2009). Their findings favour the asthenospheric flow model as origin for the anisotropy for the data set of the full data set and the northern subarea. The southern subarea allows no precise statement, most likely due to the limited event coverage. They conclude, that the anisotropy in the region is originated by a Poiseuille flow driven by the Apenninic slab rollback, likely to coincide with a Couette flow linked to the absolute plate motion. The authors argue also, that a mainly lithospheric origin for the southern subregion is to be expected, due to a dominant effect of the slab. The method applied in this paper has the potential to improve our understanding of the link between geodynamic processes and measured anisotropy. It is promising with respect to the huge data set of the previous publication, that similar results could be found here for the Central Alps indicating an asthenospheric origin. Nevertheless, there are some major and minor issues, which give rise to questions.

1) The study neglects important aspects and possibly equivalent models, that could explain the analysed data. Due to this incompleteness the conclusions drawn in this paper remain ambiguous, while a short discussion or comparison could likely clarify these open questions.

> *To introduce the new observational constraint, we juxtaposed two simple models in Fig. 2, and found that one of them matches the observations. We then introduced the new constraint into the geodynamic discussions, and addressed the main groups of models, we believe.*

2) Generally, the submitted manuscript contains some weaknesses in language and structure. While it should be revised for grammar and wording, I would also recommend reducing the section about the tectonic setting. A more focused introduction leading to the main open questions discussed in this paper would be sufficient. The paper might benefit generally from a clearer structure.

> *We tried to improve the structure of the whole manuscript following Referee #1. Just page 4 remained in the Tectonic Settings section. The Introduction got extended and includes the previous ideas of Barruol et al. (2011), Qorbani et al. (2015) and Salimbeni et al. (2018) on the relation of slab geometries and mantle flow in the region, which are of crucial importance for the understanding of the paper.*

3) As the method is based on an azimuthal variation of the splitting parameters, a more detailed discussion should be included, clarifying, why a multiple layered anisotropy can be neglected as cause for the observed azimuthal distribution. It is not sufficient to base the single layer assumption on the spatially coherent anisotropy (page 1/line 10-11, page 3/line 8). Multiple layers with individually spatial coherent anisotropy will also produce spatially coherent distribution for the measured splitting (of course with azimuthal variation at each individual station). This question remains with respect to the limited azimuthal coverage, which appears sparser in figure 4 then described in the paper (page 8/line 5). A 90° periodicity, as expected for layered anisotropy, cannot be excluded. Therefore, I would suggest showing a fit of a two-layer anisotropy at exemplary stations to allow a meaningful discussion.

> *Assuming a single-layer case of seismic anisotropy beneath the Central Alps is widely accepted. That's why we have chosen the area for our investigation. As stated in Barruol et al. (2011) referring to the distribution of shear-wave splitting parameters: "Swiss stations do not show clear evidence of backazimuthal variation of these parameters in the SKS period range (i.g., between 5 and 20s) yet the azimuthal coverage is uneven. The seismic rays are mostly incident within the NE and SW quadrant." Thus, we do not see the need for a two-layer case test. We still revised the sentences mentioned to:*
> *… a spatially coherent and relatively simple mountain-chain-parallel pattern, without large azimuthal variations per stations, …*
> *As azimuthal variations per station are comparably small (Barruol et al., 2011) a single-layer case of seismic anisotropy is likely.*
> *… accumulate mainly for backazimuths of ~ 41° - 86°, 200° and 241° - 299°.*

4) The favoured Couette-Poiseuille flow in abstract and conclusion seems contradicting to the main assumption of a single layer anisotropy resulting in an azimuthal variation of splitting parameters solely caused by the non-vertical incidence of S(K)KS-phases. A Couette-Poiseuille flow would produce a depth dependent fast axis direction resulting in azimuthal variation of phi and dt (with 90° periodicity). I agree with the first conclusion made in this paper, that the data is best explained with a Poiseuille flow model.

> *We agree in that point. The conclusion on a Couette-Poiseuille flow went too far, as despite of the worse depth resolution of S(K)KS phases, this could indeed be understood as contradicting with the single-layer assumption. We changed our initial finding in abstract and conclusion to:*
> *… the northern subarea shows indications of a planar Poiseuille flow contribution around the Alps.*
> *Instead a planar Poiseuille flow contribution …*

In the following I will list some minor suggestions for improvement

- Further topics to be discussed or mentioned

5) Page 2/Line 2: LPO of olivine is not the only origin for anisotropy (Savage 1999).

> *We mention also other minerals like orthopyroxene, clinopyroxene and garnet below, but extended the sentence now to:*
> *However, LPO of olivine is not the only origin of anisotropy (Savage, 1999). Other minerals like orthopyroxene, clinopyroxene and garnet are also anisotropic …*

6) Page 2/Line 6: What is about other depth regions and their contribution to anisotropy?

> *Other depth regions also contribute to anisotropy, but as SKS and SKKS phases are mainly sensitive to upper mantle anisotropy, we will only mention them shortly:*
> *Following previous studies, summarized in Savage (1999), the occurrence of anisotropy is not restricted to the upper mantle only. In addition, the D'' layer, as well as other depth intervals like the crust, generate anisotropic behavior. Although, the latter is related to e.g. the alignment of cracks or alternating layers of different seismic velocities.*

7) Page 2/Line 18-19: The elliptical motion is also depending on the frequency content of the core-mantle converted phase (see Rümpker & Silver 1998)

> *That is true of course. We have decided to take out that sentence though that does not really seem to be necessary for the introduction.*

8) Page 2/Line 24: This is not generally true, as there are also other effects producing anisotropy e.g. alignment of cracks in the crust or alternating layers of different seismic velocities. (Savage 1999)

> *Also here we have decided to thake out the sentence to shorten the introduction.*

9) Page 3/Line 11: It would be helpful at this point to state the contradicting arguments, and also pick up on them again in the discussions section to show how the understanding of the geodynamic processes is improved by this paper (and what arguments are ruled out by this paper).

> *As we shifted page 5 into the introduction, the models of Barruol et al. (2011), Salimbeni et al. (2018) and Qorbani et al. (2015) are now explained before. We pick up on Salimbeni et al. (2018) in our discussion and further constrain a Poiseuille flow contribution to the counter flow. The contradicting tectonics settings Kästle et al. (2019) mentioned, relate more to the slab geometry, which we cannot clarify with our new methodology. We think they are thus beyond the scope of the paper.*

10) Page 6/Line 1: This section summarizes the theory found in Löberich & Bokelmann 2020. Nevertheless, it would be important to also point out the limitations and the assumptions the theory is based on (e.g. the symmetry system).

> *Most of the assumptions and limitations are listed in the next line: Taylor-series expansion, horizontally-oriented single-layer case, orthorhombic seismic anisotropy.*

11) Page 8/Line 8: Does the number of measurements at one station have a large impact on the results or the observables dphi and ddt?

> *A larger number of measurements will be advantageous in a single-layer case. As we mention, it helps to determine the means/medians of SWS parameters per station more stably. We further added another sentence to this paragraph:*
> *A larger number of high-quality measurements per station is thus advantageous for pinpointing δφ and δΔt.*

12) Page 10/Line 26-29: The southern subarea is also characterized by less data coverage. Might that also be a reason for the different observation compared to the northern subarea?

> *Yes, that is true. We state this a bit later in the next paragraph, when comparing the results of the non-vertical-ray SWS analysis for both subareas, but as it is also a finding of the histogram, we added it to these lines:*
> *In comparison to the northern subarea, the southern subarea is less we--constrained.*

13) Page 13/ Line 12-13: That the data supports a high-temperature mechanism is an important conclusion here, as it is stated. While it is mentioned, that geological observations favour a different mechanism, are there geological observations in the area, that support this mechanism?

> *We refer here to girdle configurations often assumed by petrologists in general not specifically in the Central Alps context.*

- Suggestions for reduction

14) Page 2/Line 12-13: the comparison with optical anisotropy seems unnecessary at this point, as it is not used further to explain the properties of seismic anisotropy.

> *Ok, we removed the comparison.*

15) Page 4/Line 1: This section might be too long and detailed, as this paper aims solely to differentiate between an asthenospheric and lithospheric origin for the measured anisotropy.

> *We also see the differentiation between asthenospheric and lithospheric cause of anisotropy in the focus of our paper, but the background knowledge about the alpine geological history should not be neglected.*
> *By shifting page 5 to the Introduction following Referee #1, the Tectonic Settings were already shortened. We further tried to shrink the first paragraph, but it is already condensed.*

16) Page 12/Line 12: This statement seems not to fit to the context of the publication.

> *Ok, we shortened it to:*
> *In any case, the importance of an observation is not necessarily related to its size.*

17) Page 12/Line 23-24: It is not clear how an extended period of the AlpArray experiment is connected to the current paper.

> *This seems to be a misunderstanding. In this paragraph we consider possibilities to improve the backazimuthal coverage. As the area is already densely covered, the additional AlpArray stations would not change the overall picture drastical. Extending the time period of the permanent stations used so far would be more interesting. We rephrased the sentence to:*
> *Since the area of study is already quite well-covered with permanent broadband instruments, but the dataset so far only covers a relatively limited time duration, it can in principle be extended for a longer time period.*

18) Page 13/Line 1-6: The data set has been strongly simplified by taking the mean (and median) for the intervals around the extrema in dphi, which also coincide with ddt expected to be zero. With this simplification it is only natural not to see any further complexities, necessary to be fitted by a more complex model. With this I don't think it is necessary to point out what complexities are not considered for the modeling, as it gives no additional information for the conclusion of this paper.

> *Ok, we removed the sentence on other minerals and the pressure effect.*

- Technical comments

19) Page 2/Line 1: More generally the waves are affected by the medium they propagate through (not necessarily layers).

> *We agree, and changed it to:*
> *… properties of the medium they pass …*

20) Page2/Line 8: effect=affect

> *Due to our reply to 5) this part of the sentence was removed.*

21) Page 2/Line 18: "show up" = appears

> *We agree, and changed it to:*
> *… anisotropy appears as …*

22) Page 2/Line 19: isotropic Earth = isotropic medium

> *We agree, and changed it to:*
> *… an isotropic medium, …*

23) Page 2/Line 20: "shows a signal" – The measured time-series on the transverse component is not an independent signal. I would rather refer to the energy of the signal that occurs on the transverse component due to the splitting.

> *We agree but removed the sentence.*

24) Page 2/Line 26: "So" seems not the right word here, as resolving the foliation seems not to be related to the weak depth resolution.

> *Due to the logical inconsistency, we removed the sentence.*

25) Page 4/Line 14: "e.g." seems to be unnecessary here

> *We agree and removed it.*

26) Page 4/Line 18: "till" = until

> *We agree, and changed it to:*
> *Exhumation took place until 20 Ma …*

27) Page 6/Line 4: "SKS" and SKKS or S(K)KS

> *We agree, and changed it to:*
> *… non-vertical-ray SKS and SKKS phase arrival …*

28) Page 7/Line 5: "individuals"=individual

> *We agree, and changed it to:*
> *Distribution of high-quality individual SWS measurements …*

29) Page 8/Line 9-10: This does not seem to be a "correction". It would be better to simply describe what is done: First the mean values are subtracted from the individual values to obtain ddt and dphi. Subsequently the backazimuth is reduced by the mean of the fast axis to shift all measurements to the same reference.

> *We agree that the sentence was to complex and rephrased it to:*
> *The backazimuthal distribution of each SWS parameter is then reduced by $\bar{\varphi}$, which shifts all measurements to the same reference (second row).*

30) Page 8/Line 25: A "verification" might not be possible, but it might proof it as a most probable model, which is already very promising.

> *We agree, and changed it to:*
> *A comparison with tomography of the region can test this more.*

31) Page 10/Line 21-24: This description is slightly confusing. The statements regarding the fast axis directions don't fit in the general structure of the sentence.

> *We agree, and rephrased it to:*
> *Overall the northern subarea is characterized by longer travel times, due to the lower velocities of the asthenospheric flow below, and the shorter travel times of the southern subarea are related to the high velocities of the lithospheric slab. While φ rotates along the Alps in the north, it seems to be parallel to the anomaly in the south. Considering …*

32) Page 10/ Line 26: "transition" might be the wrong word to describe a drop of magnitude in a histogram.

> *We agree, and changed it to:*
> *… limited by sharp boundaries to …*

33) Page 13/Line 31: "shear strain does too"

> *We changed it to:*
> *… and so strain varies too.*

34) Page 14/Line 31: "Now we return"

> *We removed it.*

35) Page 14/Line 35+ Page 17/Line 24: "render" – seems to be the wrong word?

> *We agree, and replaced it with:*
> *… that transfers geodetic motions…*
> *… shear-wave splitting and vertically-integrated travel times likely suggests a contribution of the lithospheric slab.*

36) Page 17/Line 4: derived=measured; angular=azimuthal

> *We agree partly, and replaced derived with observed, as the variations are not directly measured:*
> *We have compared modeled and observed azimuthal shear-wave splitting variations …*

---

## Author Comment (AC3) · 27 May 2020

**Comments of Referee #2**

This manuscript uses the non-vertical propagation of SK(K)S phases along with previous shear wave splitting observations to model anisotropy beneath the Alps. The authors find that beneath the northern portion of their study region, anisotropy can best be modeled with a b-up olivine alignment indicative of flow in the asthenosphere. In the southern half of the study area, the slab seems to be in the way complicating the observations and producing less well-fit model results.

1) Are there any restrictions placed on the delay time errors used in the high quality splitting dataset like the 20deg restriction placed on fast direction? a. What are average errors on dts and Phis of the original splitting dataset? and the restricted dataset?

> *No, we did not further restrict the dataset for delay times errors. As in the investigated intervals Δt is expected to be 0, the conclusion unfortunately does not really benefit from delay time changes. Also, usually the determination of an amplitude, as in the case of Δt, is less precise.*
> *As we combined only the "good" quality SKS and SKKS measurements of three datasets and used the permanent stations of restricted areas, it is difficult to trace back to an "original dataset" and average errors cannot be representative for the whole shear-wave splitting datasets of Barruol et al. (2011), Qorbani et al. (2015) and Salimbeni et al. (2018). Nevertheless, a version of the dataset (not limited by the study area) before implementing a restriction on φ led to average errors of 9.04° and 0.25s in comparison to 8.14° and 0.23s afterwards.*

2) The manuscript would benefit from an expanded discussion of the 1-layer of anisotropy assumption.

a. I think a supplemental figure of baz for the stations before restricting the dataset would help. It is important to make sure that by restricting the dataset you are not removing some of the BAZ variability.

> *As stated in Barruol et al. (2011) referring to the distribution of shear-wave splitting parameters: "Swiss stations do not show clear evidence of backazimuthal variation of these parameters in the SKS period range (i.g., between 5 and 20s) yet the azimuthal coverage is uneven. The seismic rays are mostly incident within the NE and SW quadrant." Effects of multiple or dipping layers have not been resolved. We rephrased and slightly extended the discussion on single-layer anisotropy:*
> *Indeed, the Central Alps have been well-characterized by single-layer anisotropy before, as azimuthal variations per stations are comparably small (Barruol et al., 2011). Evidence for a two-layer case (90° periodicity) or a dipping layer 360° periodicity), causing upper mantel anisotropy, have not been resolved in the area.*
> *Due to the small size of the effects that we seek, they would not become visible in such a plot readily; they emerge by combining different data points.*

b. There seems to be large variances in dt, which can be indicative of layered splitting

> *The determination of Δt is usually less precise. Also, the range of the majority of observed δΔt does not necessarily require more than one layer to be considered (Fig. 4 third row). We added the following to the single-layer discussion:*
> *Strong backazimuthal changes in the delay time could be an indication of multiple anisotropic layers but considering that the determination of delay times is usually less precise, the larger scattering at individual stations for different events is also not surprising. Overall, the majority of observed δΔt per station remain in a range, comparable to the expected variations from non-vertical incidence (Fig. 4 third row).*

3) Why are there fewer BAZs represented at stations above the slab (e.g. figure 5)? Slab stations seem to have fewer 225-360deg baz splits. If this is due to dataset error restriction, it could be an indicator of layers of anisotropy. Layered splitting measurements often have larger errors at BAZs where the largest variations in phi and dt occur.

> *The smaller variation of fast axes in the southern window is intriguing. It is not due to the dataset restrictions, and neither is it easily explained by 2-layer anisotropy.*

4) How are the BAZs of 45 and 135 around which things are stacked chosen? Is it based on average fast directions? Or because of BAZ coverage? Or some initial assumption about olivine axes?

> *The intervals are chosen based on different factors, mentioned in the manuscript. They are related to the backazimuthal distribution itself and the expected variations. SWS parameters e.g. cannot be obtained for backazimuths parallel or perpendicular to φ. Excluding those backazimuthal ranges leads to the baz windows that we used.*

5) Figure 5 – change the colors of the cross section lines. It is very hard to see B-B'

> We agree and used slightly darker colors now.

6) Does the 70% of fully aligned olivine assumption make sense given the observed delay times? Wouldn't a ~100km layer with 70% alignment yield far larger delay times than the 1-2s dts that are measured?

> *Given the calculations for Fig. 2 both, the b-up and c-up case, a 100km thick layer of 70% aligned olivine lead to reasonable delay times considering the histogram in Fig. 3. However, as delay times relate to a tradeoff between the layer thickness and the strength of anisotropy, other combinations might explain the distribution similarly good.*
> *We see that this connection might have been not obvious and extended our explanations:*
> *Assuming a 100km thick layer of 70% aligned olivine with a horizontal a-axis (Nicholas and Christensen, 1987) and …*
> *… Delay times (bottom, right) range between ~0.48 - 2.88s, with an accumulation between ~0.96 - 1.92s, not unlike the expected values in Fig. 2.*

7) It would be helpful to add a discussion of how b-up and c-up olivine relate (or translate) to the more traditionally used A-, B-, C-, D-, E- type fabrics. a. A-, C- and E-type fabrics have all been proposed for the asthenospheric mantle and all have general properties where phi aligns with strain. But A-type and C-and E-types have different b-up and c-up relationships to shear strain. So how do you distinguish between them, or do you have to assume A-type?

> *We agree, in a subduction zone other olivine types are likely present. As we think that could be another factor why the southern area might deviate, we added the following to our explanations:*
> *If this area can be understood as hydrated (Giacomuzzi et al. 2011, and for further reading Hearn, 1999), serpentine, known to react highly anisotropic (see Katayama et al., 2009 or Salimbeni et al., 2018, based on Bezacier et al. 2010), and variation in the olivine type (Jung, 2009), must be considered. Indeed, evidence for different olivine types were found, e.g. B-type near Cima di Gagnone (Skemer et al., 2006); C-type around Alpe Arami (Skemer et al. 2006, based on Mockel, 1969; Buiskool Toxopeus, 1976; Frese 2002); A-, B-, and E-type at Val Malenco (Jung, 2009). However, until now we just applied our method using the San Carlos olivine of Abramson et al. (1997) in different orientations. Future studies might take the effect of different olivine types in their b-up and c-up variation on shear-wave splitting parameters into account. The diversity hinders also the comparison with surface motions derived by Sanchez et al.*

*(2018). They are slightly stronger (0.35 - 0.78mm/a), and mainly oriented towards NNW in the southern subarea (Fig. 9 bottom row).*

➢ *… effects from serpentinite and various olivine types might occur…*

8) Figs 8 and 9 and text – It would be useful to discuss and show a null test of the models. It looks to me like a flat line model would match the data just as well as the varying BAZ model.

➢ *In Fig. 4 and the northern subarea in Fig.7 (former Fig. 8) it is clear, that this Null hypothesis would be rejected by the data. Yet in the southern subarea (former Fig. 9) simple model do not seem to explain the data.*

9) P1L1 and throughout – "e.g." is used throughout the text in cases where it is not needed.

➢ *We tried to reduce it and removed it.*

10) P1L2 – "constraint" to "constrain"

➢ *We changed to:*
➢ *… constrain their nature.*

11) P4L18 – "till" to "until"

➢ *We change it to:*
➢ *… until 20 Ma …*

12) P14L33 – "first site" to "first sight"

➢ *We change it to:*
➢ *At first sight …*